# GENERATIVE MODELING
# WITH OPTIMAL TRANSPORT MAPS

**Litu Rout**
Space Applications Centre
Indian Space Research Organisation
lr@sac.isro.gov.in

**Alexander Korotin**
Skolkovo Institute of Science and Technology
Artificial Intelligence Research Institute (AIRI)
a.korotin@skoltech.ru

**Evgeny Burnaev**
Skolkovo Institute of Science and Technology
Artificial Intelligence Research Institute (AIRI)
e.burnaev@skoltech.ru

## ABSTRACT

With the discovery of Wasserstein GANs, Optimal Transport (OT) has become a powerful tool for large-scale generative modeling tasks. In these tasks, OT cost is typically used as the loss for training GANs. In contrast to this approach, we show that the OT map itself can be used as a generative model, providing comparable performance. Previous analogous approaches consider OT maps as generative models only in the latent spaces due to their poor performance in the original high-dimensional ambient space. In contrast, we apply OT maps directly in the ambient space, e.g., a space of high-dimensional images. First, we derive a min-max optimization algorithm to efficiently compute OT maps for the quadratic cost (Wasserstein-2 distance). Next, we extend the approach to the case when the input and output distributions are located in the spaces of different dimensions and derive error bounds for the computed OT map. We evaluate the algorithm on image generation and unpaired image restoration tasks. In particular, we consider denoising, colorization, and inpainting, where the optimality of the restoration map is a desired attribute, since the output (restored) image is expected to be close to the input (degraded) one.

## 1 INTRODUCTION

Since the discovery of Generative Adversarial Networks (GANs, Goodfellow et al. (2014)), there has been a surge in generative modeling (Radford et al., 2016; Arjovsky et al., 2017; Brock et al., 2019; Karras et al., 2019). In the past few years, Optimal Transport (OT, Villani (2008)) theory has been pivotal in addressing important issues of generative models. In particular, the usage of Wasserstein distance has improved diversity (Arjovsky et al., 2017; Gulrajani et al., 2017), convergence (Sanjabi et al., 2018), and stability (Miyato et al., 2018; Kim et al., 2021) of GANs.

Generative models based on OT can be split into two classes depending on what OT is used for. First, the **optimal transport cost serves as the loss** for generative models, see Figure 1a. This is the most prevalent class of methods which includes WGAN (Arjovsky et al., 2017) and its modifications: WGAN-GP (Gulrajani et al., 2017), WGAN-LP (Petzka et al., 2018), and WGAN-QC (Liu et al., 2019). Second, the **optimal transport map is used as a generative model** itself, see Figure 1b. Such approaches include LSOT (Seguy et al., 2018), AE-OT (An et al., 2020a), ICNN-OT (Makkuva et al., 2020), W2GN (Korotin et al., 2021a). Models of the first class have been well-studied, but limited attention has been paid to the second class. Existing approaches of the second class primarily consider **OT maps in latent spaces** of pre-trained autoencoders (AE), see Figure 3. The performance of such generative models depends on the underlying AEs, in which decoding transformations are often not accurate; as a result this deficiency limits practical applications in high-dimensional ambient spaces. For this reason, using OT in the latent space does not necessarily guarantee superior performance in generative modeling.

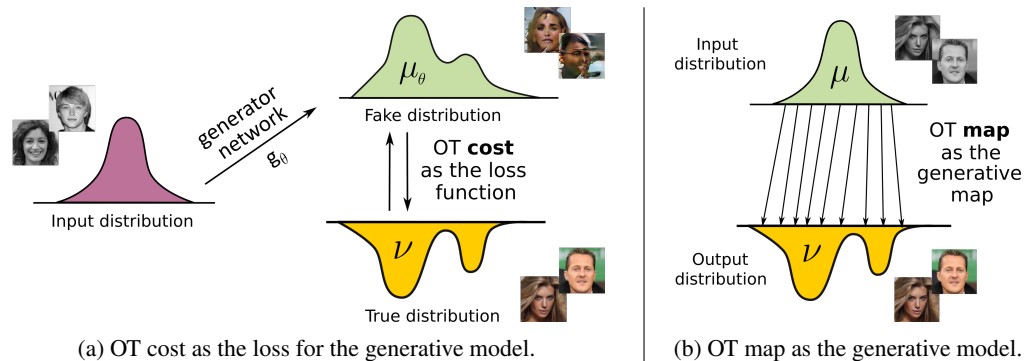

(a) OT cost as the loss for the generative model.    (b) OT map as the generative model.

Figure 1: Two existing approaches to use optimal transport in generative models.

The focus of our paper is the second class of OT-based models using OT map as the generative map. Finding an optimal mapping is motivated by its ability to preserve specific attributes of the input samples, a desired property in unpaired learning. For example, in unpaired image-to-image translation, the learner has to fit a map between two data distributions which preserves the image content. CycleGAN-based models (Zhu et al., 2017) are widely used for this purpose. However, they typically have complex optimization objectives consisting of several losses (Amodio & Krishnaswamy, 2019; Lu et al., 2019) in order to make the fitted map preserve the required attributes.

**The main contributions of this paper are as follows:**

1. We propose an end-to-end algorithm (§4.3) to fit OT maps for the quadratic cost (Wasserstein-2 distance) between distributions located on the spaces of equal dimensions (§4.1) and extend the method to unequal dimensions as well (§4.2). We prove error bounds for the method (§4.4).

2. We demonstrate large-scale applications of OT maps in popular computer vision tasks. We consider image generation (§5.1) and unpaired image restoration (§5.2) tasks.

Our strict OT-based framework allows the theoretical analysis of the recovered transport map. The OT map obtained by our method can be directly used in large-scale computer vision problems which is in high contrast to previous related methods relying on autoencoders and OT maps in the latent space. Importantly, the performance and computational complexity of our method is comparable to OT-based generative models using OT cost as the loss.

**Notations.** In what follows, $\mathcal{X}$ and $\mathcal{Y}$ are two complete metric spaces, $\mu(x)$ and $\nu(y)$ are probability distributions on $\mathcal{X}$ and $\mathcal{Y}$, respectively. For a measurable map $T : \mathcal{X} \to \mathcal{Y}$, $T_{\#}\mu$ denotes the pushforward distribution of $\mu$, i.e., the distribution for which any measurable set $E \subset \mathcal{Y}$ satisfies $T_{\#}\mu(E) = \mu(T^{-1}(E))$. For a vector $x$, $\|x\|$ denotes its Euclidean norm. We use $\langle x, y \rangle$ to denote the inner product of vectors $x$ and $y$. We use $\Pi(\mu, \nu)$ to denote the set of joint probability distributions on $\mathcal{X} \times \mathcal{Y}$ whose marginals are $\mu$ and $\nu$, respectively (couplings). For a function $f : \mathbb{R}^D \to \mathbb{R} \cup \{\pm\infty\}$ its Legendre–Fenchel transform (the convex conjugate) is $\overline{f}(y) = \sup_{x \in \mathbb{R}^D} \{\langle x, y \rangle - f(x)\}$. It is convex, even if $f$ is not.

## 2 BACKGROUND ON OPTIMAL TRANSPORT

Consider a cost of transportation, $c : \mathcal{X} \times \mathcal{Y} \to \mathbb{R}$ defined over the product space of $\mathcal{X}$ and $\mathcal{Y}$.

**Monge's Formulation.** The *optimal transport cost* between $\mu$ and $\nu$ for ground cost $c(\cdot, \cdot)$ is

$$\text{Cost}(\mu, \nu) \overset{\text{def}}{=} \inf_{T_{\#}\mu=\nu} \int_{\mathcal{X}} c(x, T(x)) \, d\mu(x), \quad (1)$$

where the infimum is taken over all measurable maps $T : \mathcal{X} \to \mathcal{Y}$ pushing $\mu$ to $\nu$, see Figure 2. The map $T^*$ on which the infimum in (1) is attained is called the *optimal transport*

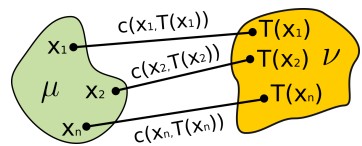

Figure 2: Monge's OT.

*map*. Monge's formulation does not allow splitting. For example, when $\mu$ is a Dirac distribution and $\nu$ is a non-Dirac distribution, the feasible set of equation (1) is empty.

**Kantorovich's Relaxation.** Instead of asking to which particular point $y \in \mathcal{Y}$ should all the probability mass of $x$ be moved, Kantorovich (1948) asks how the mass of $x$ should be distributed among all $y \in \mathcal{Y}$. Formally, a transport coupling replaces a transport map; the OT cost is given by:

$$\text{Cost}(\mu, \nu) \stackrel{\text{def}}{=} \inf_{\pi \in \Pi(\mu,\nu)} \int_{\mathcal{X} \times \mathcal{Y}} c(x, y) \, d\pi(x, y), \tag{2}$$

where the infimum is taken over all couplings $\pi \in \Pi(\mu, \nu)$ of $\mu$ and $\nu$. The coupling $\pi^*$ attaining the infimum of (2) is called the *optimal transport plan*. Unlike the formulation of (1), the formulation of (2) is well-posed, and with mild assumptions on spaces $\mathcal{X}, \mathcal{Y}$ and ground cost $c(\cdot, \cdot)$, the minimizer $\pi^*$ of (2) always exists (Villani, 2008, Theorem 4.1). In particular, if $\pi^*$ is deterministic, i.e., $\pi^* = [\text{id}_{\mathcal{X}}, T^*]_{\#}\mu$ for some $T^* : \mathcal{X} \to \mathcal{Y}$, then $T^*$ minimizes (1).

**Duality.** The dual form of (2) is given by (Kantorovich, 1948):

$$\text{Cost}(\mu, \nu) = \sup_{(u,v)} \left\{ \int_{\mathcal{X}} u(x) d\mu(x) + \int_{\mathcal{Y}} v(y) d\nu(y) \colon u(x) + v(y) \leq c(x, y) \right\}, \tag{3}$$

with $u \in L^1(\mu)$, $v \in L^1(\nu)$ called Kantorovich *potentials*. For $u : \mathcal{X} \to \mathbb{R}$ and $v : \mathcal{Y} \to \mathbb{R}$ define their $c$-transforms by $u^c(y) = \inf_{x \in \mathcal{X}} \{c(x, y) - u(x)\}$ and $v^c(x) = \inf_{y \in \mathcal{Y}} \{c(x, y) - v(y)\}$ respectively. Using $c$-transform, (3) is reformulated as (Villani, 2008, §5)

$$\text{Cost}(\mu, \nu) = \sup_v \left\{ \int_{\mathcal{X}} v^c(x) d\mu(x) + \int_{\mathcal{Y}} v(y) d\nu(y) \right\} = \sup_u \left\{ \int_{\mathcal{X}} u(x) d\mu(x) + \int_{\mathcal{Y}} u^c(y) d\nu(y) \right\}. \tag{4}$$

**Primal-dual relationship.** For certain ground costs $c(\cdot, \cdot)$, the primal solution $T^*$ of (1) can be recovered from the dual solution $u^*$ of (3). For example, if $\mathcal{X} = \mathcal{Y} = \mathbb{R}^D$, $c(x, y) = h(x - y)$ with strictly convex $h : \mathbb{R}^D \to \mathbb{R}$ and $\mu$ is absolutely continuous supported on the compact set, then

$$T^*(x) = x - (\nabla h)^{-1} (\nabla u^*(x)), \tag{5}$$

see (Santambrogio, 2015, Theorem 1.17). For general costs, see (Villani, 2008, Theorem 10.28).

## 3 OPTIMAL TRANSPORT IN GENERATIVE MODELS

**OPTIMAL TRANSPORT COST AS THE LOSS** (Figure 1a). Starting with the works of Arjovsky & Bottou (2017); Arjovsky et al. (2017), the usage of OT cost as the loss has become a major way to apply OT for generative modeling. In this setting, given data distribution $\nu$ and fake distribution $\mu_\theta$, the goal is to minimize $\text{Cost}(\mu_\theta, \nu)$ w.r.t. the parameters $\theta$. Typically, $\mu_\theta$ is a pushforward distribution of some given distribution, e.g., $\mathcal{N}(0, I)$, via generator network $G_\theta$.

The *Wasserstein-1* distance ($\mathcal{W}_1$), i.e., the transport cost for ground cost $c(x, y) = \|x - y\|$, is the most practically prevalent example of such a loss. Models based on this loss are known as Wasserstein GANs (WGANs). They estimate $\mathcal{W}_1(\mu_\theta, \nu)$ based on the dual form as given by (4). For $\mathcal{W}_1$, the optimal potentials $u^*, v^*$ of (4) satisfy $u^* = -v^*$ where $u^*$ is a 1-Lipschitz function (Villani, 2008, Case 5.16). As a result, to compute $\mathcal{W}_1$, one needs to optimize the following simplified form:

$$\mathcal{W}_1(\mu_\theta, \nu) = \sup_{\|u\|_L \leq 1} \left\{ \int_{\mathcal{X}} u(x) d\mu_\theta(x) - \int_{\mathcal{Y}} u(y) d\nu(y) \right\}. \tag{6}$$

In WGANs, the potential $u$ is called the *discriminator*. Optimization of (6) reduces constrained optimization of (4) with two potentials $u, v$ to optimization of only one discriminator $u$. In practice, enforcing the Lipschitz constraint on $u$ is challenging. Most methods to do this are regularization-based, e.g., they use gradient penalty (Gulrajani et al., 2017, WGAN-GP) and Lipschitz penalty (Petzka et al., 2018, WGAN-LP). Other methods enforce Lipschitz property via incorporating certain hard restrictions on the discriminator's architecture (Anil et al., 2019; Tanielian & Biau, 2021).

*General transport costs* (other than $\mathcal{W}_1$) can also be used as the loss for generative models. They are less popular since they do not have a dual form reducing to a single potential function similar to (6) for $\mathcal{W}_1$. Consequently, the challenging estimation of the $c$-transform $u^c$ is needed. To

avoid this, Sanjabi et al. (2018) consider the dual form of (3) with two potentials $u, v$ instead form (4) with one $u$ and softly enforce the condition $u(x) + v(y) \leq c(x, y)$ via entropy or quadratic regularization. Nhan Dam et al. (2019) use the dual form of (4) and amortized optimization to compute $u^c$ via an additional neural network. Both methods work for general $c(\cdot, \cdot)$, though the authors test them for $c(x, y) = \|x - y\|$ only, i.e., $\mathcal{W}_1$ distance. Mallasto et al. (2019) propose a fast way to approximate the $c$-transform and test the approach (WGAN-$(q, p)$) with several costs, in particular, the *Wasserstein-2* distance ($\mathcal{W}_2$), i.e., the transport cost for the quadratic ground cost $c(x, y) = \frac{1}{2}\|x - y\|^2$. Specifically for $\mathcal{W}_2$, Liu et al. (2019) approximate the $c$-transform via a linear program (WGAN-QC).

A fruitful branch of OT-based losses for generative models comes from modified versions of OT cost, such as Sinkhorn (Genevay et al., 2018), sliced (Deshpande et al., 2018) and minibatch (Fatras et al., 2019) OT distances. They typically have lower sample complexity than usual OT and can be accurately estimated from random mini-batches without using dual forms such as (3). In practice, these approaches usually learn the ground OT cost $c(\cdot, \cdot)$.

The aforementioned methods use OT cost in the ambient space to train GANs. There also exist approaches using OT cost in the latent space. For example, Tolstikhin et al. (2017); Patrini et al. (2020) use OT cost between encoded data and a given distribution as an additional term to reconstruction loss for training an AE. As the result, AE's latent distribution becomes close to the given one.

**OPTIMAL TRANSPORT MAP AS THE GENERATIVE MAP** (Figure 1b). Methods to compute the OT map (plan) are less common in comparison to those computing the cost. Recovering the map from the primal form (1) or (2) usually yields complex optimization objectives containing several adversarial terms (Xie et al., 2019; Liu et al., 2021; Lu et al., 2020). Such procedures require careful hyperparameter choice. This needs to be addressed before using these methods in practice.

Primal-dual relationship (§2) makes it possible to recover the OT map via solving the dual form (3). Dual-form based methods primarily consider $\mathcal{W}_2$ cost due to its nice theoretical properties and relation to convex functions (Brenier, 1991). In the semi-discrete case ($\mu$ is continuous, $\nu$ is discrete), An et al. (2020a) and Lei et al. (2019) compute the dual potential and the OT map by using the Alexandrov theory and convex geometry. For the continuous case, Seguy et al. (2018) use the entropy (quadratic) regularization to recover the dual potentials and extract OT map from them via the barycenteric projection. Taghvaei & Jalali (2019), Makkuva et al. (2020), Korotin et al. (2021a) employ input-convex neural networks (ICNNs, see Amos et al. (2017)) to parametrize potentials in the dual problem and recover OT maps by using their gradients.

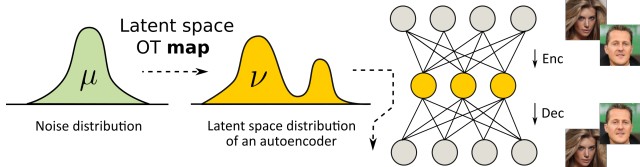

Figure 3: The existing most prevalent approach to use OT maps in generative models.

The aforementioned dual form methods compute OT maps in **LATENT SPACES** for problems such as domain adaptation and latent space mass transport, see Figure 3. OT maps in high-dimensional ambient spaces, e.g., natural images, are usually not considered. Recent evaluation of continuous OT methods for $\mathcal{W}_2$ (Korotin et al., 2021b) reveals their crucial limitations, which negatively affect their scalability, such as poor expressiveness of ICNN architectures or bias due to regularization.

## 4 END-TO-END SOLUTION TO LEARN OPTIMAL MAPS

### 4.1 EQUAL DIMENSIONS OF INPUT AND OUTPUT DISTRIBUTIONS

In this section, we use $\mathcal{X} = \mathcal{Y} = \mathbb{R}^D$ and consider the Wasserstein-2 distance ($\mathcal{W}_2$), i.e., the optimal transport for the quadratic ground cost $c(x, y) = \frac{1}{2}\|x - y\|^2$. We use the dual form (4) to derive a saddle point problem the solution of which yields the OT map $T^*$. We consider distributions $\mu, \nu$ with finite second moments. We assume that for distributions $\mu, \nu$ in view there *exists* a *unique* OT plan $\pi^*$ minimizing (3) and it is deterministic, i.e., $\pi^* = [\mathrm{id}_{\mathbb{R}^D}, T^*]_{\#}\mu$. Here $T^*$ is an OT map

which minimizes (1). Previous related works (Makkuva et al., 2020; Korotin et al., 2021a) assumed the absolute continuity of $\mu$, which implied the existence and uniqueness of $T^*$ (Brenier, 1991).

Let $\psi(y) \stackrel{\text{def}}{=} \frac{1}{2}\|y\|^2 - v(y)$, where $v$ is the potential of (4). Note that

$$v^c(x) = \inf_{y \in \mathbb{R}^D} \left\{ \frac{1}{2}\|x - y\|^2 - v(y) \right\} = \frac{1}{2}\|x\|^2 - \sup_{y \in \mathbb{R}^D} \left\{ \langle x, y \rangle - \psi(y) \right\} = \frac{1}{2}\|x\|^2 - \overline{\psi}(x). \quad (7)$$

Therefore, (4) is equivalent to

$$\mathcal{W}_2^2(\mu, \nu) = \int_{\mathcal{X}} \frac{\|x\|^2}{2} d\mu(x) + \int_{\mathcal{Y}} \frac{\|y\|^2}{2} d\nu(x) + \sup_{\psi} \left\{ -\int_{\mathcal{X}} \overline{\psi}(x) d\mu(x) - \int_{\mathcal{Y}} \psi(y) d\nu(y) \right\} = \quad (8)$$

$$\text{Constant}(\mu, \nu) - \inf_{\psi} \left\{ \int_{\mathcal{X}} \overline{\psi}(x) d\mu(x) + \int_{\mathcal{Y}} \psi(y) d\nu(y) \right\} = \quad (9)$$

$$\text{Constant}(\mu, \nu) - \inf_{\psi} \left\{ \int_{\mathcal{X}} \sup_{y \in \mathbb{R}^D} \left\{ \langle x, y \rangle - \psi(y) \right\} d\mu(x) + \int_{\mathcal{Y}} \psi(y) d\nu(y) \right\} = \quad (10)$$

$$\text{Constant}(\mu, \nu) - \inf_{\psi} \left\{ \sup_{T} \int_{\mathcal{X}} \left\{ \langle x, T(x) \rangle - \psi\bigl(T(x)\bigr) \right\} d\mu(x) + \int_{\mathcal{Y}} \psi(y) d\nu(y) \right\} \quad (11)$$

where between lines (10) and (11) we replace the optimization over $y \in \mathbb{R}^D$ with the equivalent optimization over functions $T : \mathbb{R}^D \to \mathbb{R}^D$. The equivalence follows from the interchange between the integral and the supremum (Rockafellar, 1976, Theorem 3A). We also provide an independent proof of equivalence specializing Rockafellar's interchange theorem in Appendix A.1. Thanks to the following lemma, we may solve saddle point problem (11) and obtain the OT map $T^*$ from its solution $(\psi^*, T^*)$.

**Lemma 4.1.** *Let $T^*$ be the OT map from $\mu$ to $\nu$. Then, for every optimal potential $\psi^*$,*

$$T^* \in \arg\sup_{T} \int_{\mathcal{X}} \left\{ \langle x, T(x) \rangle - \psi^*\bigl(T(x)\bigr) \right\} d\mu(x). \quad (12)$$

We prove Lemma 4.1 in Appendix A.2. For general $\mu, \nu$ the $\arg\sup_T$ set for optimal $\psi^*$ might contain not only OT map $T^*$, but other functions as well. Working with real-world data in experiments (§5.2), we observe that despite this issue, optimization (11) still recovers $T^*$.

**Relation to previous works.** The use of the function $T$ to approximate the $c$-transform was proposed by Nhan Dam et al. (2019) to estimate the Wasserstein loss in WGANs. For $\mathcal{W}_2$, the fact that $T^*$ is an OT map was used by Makkuva et al. (2020); Korotin et al. (2021a) who primarily assumed continuous $\mu, \nu$ and reduced (11) to convex $\psi$ and $T = \nabla\phi$ for convex $\phi$. Issues with non-uniqueness of solution of (12) were softened, but using ICNNs to parametrize $\psi$ became necessary.

Korotin et al. (2021b) demonstrated that ICNNs negatively affect practical performance of OT and tested an unconstrained formulation similar to (11). As per the evaluation, it provided the best empirical performance (Korotin et al., 2021b, §4.5). The method $\lfloor$MM:R$\rfloor$ they consider parametrizes $\frac{1}{2}\|\cdot\|^2 - \psi(\cdot)$ by a neural network, while we directly parametrize $\psi(\cdot)$ by a neural network (§4.3).

Recent work by Fan et al. (2021) exploits formulation similar to (11) for general costs $c(\cdot, \cdot)$. While their formulation leads to a max-min scheme with general costs (Fan et al., 2021, Theorem 3), our approach gives rise to a min-max method for quadratic cost. In particular, we extend the formulation to learn OT maps between distributions in spaces with unequal dimensions, see the next subsection.

## 4.2 Unequal Dimensions of Input and Output Distributions

Consider the case when $\mathcal{X} = \mathbb{R}^H$ and $\mathcal{Y} = \mathbb{R}^D$ have different dimensions, i.e., $H \neq D$. In order to map the probability distribution $\mu$ to $\nu$, a straightforward solution is to embed $\mathcal{X}$ to $\mathcal{Y}$ via *some* $Q : \mathcal{X} \to \mathcal{Y}$ and then to fit the OT map between $Q_{\#}\mu$ and $\nu$ for the quadratic cost on $\mathcal{Y} = \mathbb{R}^D$. In this case, the optimization objective becomes

$$\inf_{\psi} \sup_{T} \left\{ \int_{\mathcal{X}} \left\{ \langle Q(x), T(Q(x)) \rangle - \psi\bigl(T(Q(x))\bigr) \right\} d\mu(x) + \int_{\mathcal{Y}} \psi(y) d\nu(y) \right\} \quad (13)$$

with the optimal $T^*$ recovering the OT map from $Q_{\#}\mu$ to $\nu$. For equal dimensions $H = D$ and the identity embedding $Q(x) \equiv x$, expression (13) reduces to optimization (11) up to a constant.

Instead of optimizing (13) over functions $T : Q(\mathcal{X}) \to \mathcal{Y}$, we propose to consider optimization directly over generative mappings $G : \mathcal{X} \to \mathcal{Y}$:

$$\mathcal{L}(\psi, G) \overset{\text{def}}{=} \inf_{\psi} \sup_{G} \left\{ \int_{\mathcal{X}} \left\{ \langle Q(x), G(x) \rangle - \psi\big(G(x)\big) \right\} d\mu(x) + \int_{\mathcal{Y}} \psi(y) d\nu(y) \right\} \qquad (14)$$

Our following lemma establishes connections between (14) and OT with unequal dimensions:

**Lemma 4.2.** *Assume that exists a unique OT plan between $Q_{\#}\mu$ and $\nu$ and it is deterministic, i.e., $[id_{\mathbb{R}^D}, T^*]_{\#}(Q_{\#}\mu)$. Then $G^*(x) = T^*\big(Q(x)\big)$ is the OT map between $\mu$ and $\nu$ for the $Q$-**embedded quadratic cost** $c(x, y) = \frac{1}{2}\|Q(x) - y\|^2$. Moreover, for every optimal potential $\psi^*$ of problem (14),*

$$G^* \in \arg\sup_{G} \int_{\mathcal{X}} \left\{ \langle Q(x), G(x) \rangle - \psi^*\big(G(x)\big) \right\} d\mu(x). \qquad (15)$$

We prove Lemma 4.2 in Appendix A.3 and schematically present its idea in Figure 4. Analogously to Lemma 4.1, it provides a way to compute the OT map $G^*$ for the $Q$-embedded quadratic cost between distributions $\mu$ and $\nu$ by solving the saddle point problem (14). Note the situation with non-uniqueness of $\arg\sup_G$ is similar to §4.1.

**Relation to previous works.** In practice, learning OT maps directly between spaces of unequal dimensions was considered in the work by (Fan et al., 2021, §5.2) but only on toy examples. We demonstrate that our method works well in large-scale generative modeling tasks (§5.1). Theoretical properties of OT maps for embedded costs are studied, e.g., in (Pass, 2010; McCann & Pass, 2020).

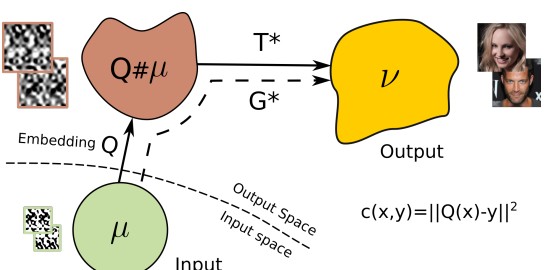

Figure 4: The scheme of our approach for learning OT maps between unequal dimensions. In the figure, the setup of §5.1 is shown: $\mu$ is a noise, $Q$ is the bicubic upscaling, $\nu$ is a distribution of images.

### 4.3 PRACTICAL ASPECTS AND OPTIMIZATION PROCEDURE

To optimize functional (14), we approximate $G : \mathbb{R}^H \to \mathbb{R}^D$ and $\psi : \mathbb{R}^D \to \mathbb{R}$ with neural networks $G_\theta, \psi_\omega$ and optimize their parameters via stochastic gradient descent-ascent (SGDA) by using mini-batches from $\mu, \nu$. The practical optimization procedure is given in Algorithm 1 below. Following the usual practice in GANs, we add a small penalty (§B.3) on potential $\psi_\omega$ for better stability. The penalty is not included in Algorithm 1 to keep it simple.

**Relation to previous works.** WGAN by Arjovsky & Bottou (2017) uses $\mathcal{W}_1$ as the loss to update the generator while we solve a diferent task — we fit the generator $G$ to be the OT map for $Q$-embedded quadratic cost. Despite this, our Algorithm 1 has similarities with WGAN's training. The update of $\psi$ (line 4) coincides with discriminator's update in WGAN. The update of generator $G$ (line 8) differs from WGAN's update by the term $-\langle Q(\cdot), G_\theta(\cdot) \rangle$. Besides, in WGAN the optimization is $\inf_G \sup_D$. We have $\inf_\psi \sup_G$, i.e., the generator in our case is the solution of the inner problem.

### 4.4 ERROR ANALYSIS

Given a pair $(\hat{\psi}, \hat{G})$ approximately solving (14), a natural question to ask is how good is the recovered OT map $\hat{G}$. In this subsection, we provide a bound on the difference between $G^*$ and $\hat{G}$ based on the duality gaps for solving outer and inner optimization problems.

In (8), and, as the result, in (10), (11), (13), (14), it is enough to consider optimization over *convex* functions $\psi$, see (Villani, 2008, Case 5.17). Our theorem below assumes the convexity of $\hat{\psi}$ although it might not hold in practice since in practice $\hat{\psi}$ is a neural network.

---

**Algorithm 1:** Learning the optimal transport map between unequal dimensions.

---

**Input** : Input distribution $\mu$ on $\mathcal{X} = \mathbb{R}^H$; output distribution $\nu$ on $\mathcal{Y} = \mathbb{R}^D$;
    generator network $G_\theta : \mathbb{R}^H \to \mathbb{R}^D$; potential network $\psi_\omega : \mathbb{R}^D \to \mathbb{R}$;
    number of iterations per network: $K_G, K_\psi$; embedding $Q : \mathcal{X} \to \mathcal{Y}$;

**Output:** Trained generator $G_\theta$ representing OT map from $\mu$ to $\nu$;

1 **repeat**
2   **for** $k_\psi = 1$ to $K_\psi$ **do**
3    Draw batch $X \sim \mu$ and $Y \sim \nu$;
4    $\mathcal{L}_\psi \leftarrow \frac{1}{|Y|} \sum_{y \in Y} \psi_\omega(y) - \frac{1}{|X|} \sum_{x \in X} \psi_\omega \big( G_\theta(x) \big)$;
5    Update $\omega$ by using $\frac{\partial \mathcal{L}_\psi}{\partial \omega}$ to minimize $\mathcal{L}_\psi$;
6   **for** $k_G = 1$ to $K_G$ **do**
7    Draw batch $X \sim \mu$;
8    $\mathcal{L}_G \leftarrow \frac{1}{|X|} \sum_{x \in X} \big[ \psi \big( G(x) \big) - \langle Q(x), G_\theta(x) \rangle \big]$;
9    Update $\theta$ by using $\frac{\partial \mathcal{L}_G}{\partial \theta}$ to minimize $\mathcal{L}_G$;
10 **until** not converged;
11 **return** $G_\theta$

---

**Theorem 4.3.** *Assume that there exists a unique deterministic OT plan for Q-embedded quadratic cost between $\mu$ and $\nu$, i.e., $\pi^* = [id_{\mathbb{R}^H}, G^*]_{\#}\mu$ for $G^* : \mathbb{R}^H \to \mathbb{R}^D$. Assume that $\hat{\psi}$ is $\beta$-strongly convex ($\beta > 0$) and $\hat{G} : \mathbb{R}^H \to \mathbb{R}^D$. Define*

$$\epsilon_1 = \sup_G \mathcal{L}(\hat{\psi}, G) - \mathcal{L}(\hat{\psi}, \hat{G}) \qquad and \qquad \epsilon_2 = \sup_G \mathcal{L}(\hat{\psi}, G) - \inf_\psi \sup_G \mathcal{L}(\psi, G)$$

*Then the following bound holds true for the OT map $G^*$ from $\mu$ to $\nu$:*

$$\frac{\text{FID}(\hat{G}_{\#}\mu, \nu)}{L^2} \leq 2 \cdot \mathcal{W}_2^2(\hat{G}_{\#}\mu, \nu) \leq \int_{\mathcal{X}} \|\hat{G}(x) - G^*(x)\|^2 d\mu(x) \leq \frac{2}{\beta}(\sqrt{\epsilon_1} + \sqrt{\epsilon_2})^2, \quad (16)$$

*where FID is the Fréchet inception distance (Heusel et al., 2017) and $L$ is the Lipschitz constant of the feature extractor of the pre-trained InceptionV3 neural network (Szegedy et al., 2016).*

We prove Theorem 4.3 in Appendix A.4. The duality gaps upper bound $L^2(\mu)$ norm between computed $\hat{G}$ and true $G^*$ maps, and the $\mathcal{W}_2^2$ between true $\nu$ and generated (fake) distribution $\hat{G}_{\#}\mu$. Consequently, they upper bound FID between data $\nu$ and fake (generated) $\hat{G}_{\#}\mu$ distributions.

**Relation to previous works.** Makkuva et al. (2020); Korotin et al. (2021a) prove related bounds for $\mathcal{W}_2$ with $\mu, \nu$ located on the spaces of the same dimension. Our result holds for different dimensions.

## 5 EXPERIMENTS

We evaluate our algorithm in generative modeling of the data distribution from a noise (§5.1) and unpaired image restoration task (§5.2). Technical details are given in Appendix B. Additionally, in Appendix B.4 we test our method on toy 2D datasets and evaluate it on the Wasserstein-2 benchmark (Korotin et al., 2021b) in Appendix B.2. The code is in the supplementary material.

### 5.1 MODELING DATA DISTRIBUTION FROM NOISE DISTRIBUTION

In this subsection, $\mu$ is a 192-dimensional normal noise and $\nu$ the high-dimensional data distribution.

Let the images from $\nu$ be of size $w \times h$ with $c$ channels. As the embedding $Q : \mathcal{X} \to \mathcal{Y}$ we use a *naive* upscaling of a noise. For $x \in \mathbb{R}^{192}$ we represent it as 3-channel $8 \times 8$ image and bicubically upscale it to the size $w \times h$ of data images from $\nu$. For grayscale images drawn from $\nu$, we stack $c$ copies over channel dimension.

We test our method on MNIST $32 \times 32$ (LeCun et al., 1998), CIFAR10 $32 \times 32$ (Krizhevsky et al., 2009), and CelebA $64 \times 64$ (Liu et al., 2015) image datasets. In Figure 5, we show random samples

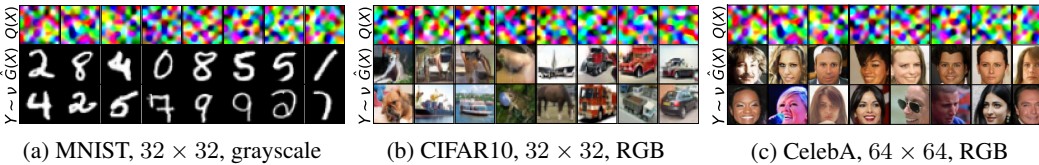

(a) MNIST, $32 \times 32$, grayscale     (b) CIFAR10, $32 \times 32$, RGB     (c) CelebA, $64 \times 64$, RGB

Figure 5: Randomly generated MNIST, CIFAR10, and CelebA samples by our method (OTM).

Table 1: Results on CIFAR10 dataset.

| Model | Related Work | Inception ↑ | FID ↓ |
|---|---|---|---|
| NVAE | Vahdat & Kautz (2020) | - | 51.71 |
| PixelIQN | Ostrovski et al. (2018) | 5.29 | 49.46 |
| EBM | Du & Mordatch (2019) | 6.02 | 40.58 |
| DCGAN | Radford et al. (2016) | 6.64±0.14 | 37.70 |
| NCSN | Song & Ermon (2019) | 8.87±0.12 | 25.32 |
| NCP-VAE | Aneja et al. (2021) | - | 24.08 |
| WGAN | Arjovsky et al. (2017) | - | 55.2 |
| WGAN-GP | Gulrajani et al. (2017) | 6.49±0.09 | 39.40 |
| 3P-WGAN | Nhan Dam et al. (2019) | 7.38 ± 0.08 | 28.8 |
| AE-OT | An et al. (2020a) | - | 28.5 |
| AE-OT-GAN | An et al. (2020b) | - | 17.1 |
| OTM | Ours | 7.42±0.06 | 21.78 |

Table 2: Results on CelebA dataset.

| Model | Related Work | FID ↓ |
|---|---|---|
| DCGAN | Radford et al. (2016) | 52.0 |
| DRAGAN | Kodali et al. (2017) | 42.3 |
| BEGAN | Berthelot et al. (2017) | 38.9 |
| NVAE | Vahdat & Kautz (2020) | 13.4 |
| NCP-VAE | Aneja et al. (2021) | 5.2 |
| WGAN | Arjovsky et al. (2017) | 41.3 |
| WGAN-GP | Gulrajani et al. (2017) | 30.0 |
| WGAN-QC | Liu et al. (2019) | 12.9 |
| AE-OT | An et al. (2020a) | 28.6 |
| AE-OT-GAN | An et al. (2020b) | 7.8 |
| OTM | Ours | 6.5 |

generated by our approach, namely **Optimal Transport Modeling** (OTM). To quantify the results, in Tables 1 and 2 we give the inception (Salimans et al., 2016) and FID (Heusel et al., 2017) scores of generated samples. Similar to (Song & Ermon, 2019, Appendix B.2), we compute them on 50K real and generated samples. Additionally, in Appendix B.4, we test our method on $128 \times 128$ CelebA faces. We provide qualitative results (images of generated faces) in Figure 11.

For comparison, we include the scores of existing generative models of three types: (1) OT map as the generative model; (2) OT cost as the loss; (3) not OT-based. Note that models of the first type compute OT in the *latent space* of an autoencoder in contrast to our approach. According to our evaluation, the performance of our method is better or comparable to existing alternatives.

## 5.2 UNPAIRED IMAGE RESTORATION

In this subsection, we consider unpaired image restoration tasks on CelebA faces dataset. In this case, the input distribution $\mu$ consists of degraded images, while $\nu$ are clean images. In all the cases, embedding $Q$ is a straightforward identity embedding $Q(x) \equiv x$.

In image restoration, optimality of the restoration map is desired since the output (restored) image is expected to be close to the input (degraded) one minimizing the transport cost. Note that GANs do not seek for an optimal mapping. However, in practice, due to implicit inductive biases such as convolutional architectures, GANs still tend to fit low transport cost maps (Bézenac et al., 2021).

The experimental setup is shown in Figure 6. We split the dataset in 3 parts A, B, C containing 90K, 90K, 22K samples respectively. To each image we apply the degradation transform (decolorization, noising or occlusion) and obtain the degraded dataset containing of 3 respective parts A, B, C. For *unpaired* training we use part A of degraded and part B of clean images. For testing, we use parts C.

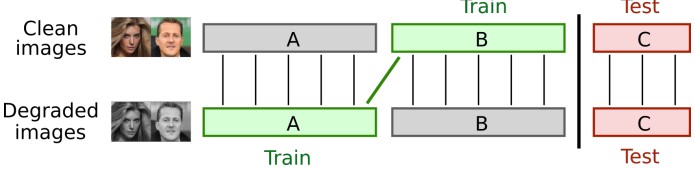

Figure 6: The training/testing scheme that we use for unpaired restoration tasks.

To quantify the results we compute FID of restored images w.r.t. clean images of part C. The scores for denoising, inpainting and colorization are given in Table 3, details of each experiment and qualitative results are given below.

As a baseline, we include WGAN-GP. For a fair comparison, we fit it using *exactly* the same hyperparameters as in our method OTM-GP. This is possible due to the similarities between our method and WGAN-GP's training procedure, see discussion in §4.3. In OTM, there is no GP (§B.3).

| Model | Denoising | Colorization | Inpainting |
|---|---|---|---|
| Input | 166.59 | 32.12 | 47.65 |
| WGAN-GP | 25.49 | 7.75 | 16.51 |
| OTM-GP (ours) | 10.95 | 5.66 | 9.96 |
| OTM (ours) | 5.92 | 5.65 | 8.13 |

Table 3: FID↓ on test part C in image restoration experiments.

**Denoising**. To create noisy images, we add white normal noise with $\sigma = 0.3$ to each pixel. Figure 7 illustrates image denoising using our OTM approach on the test part of the dataset. We show additional qualitative results for varying $\sigma$ in Figure 15 of §(B.4).

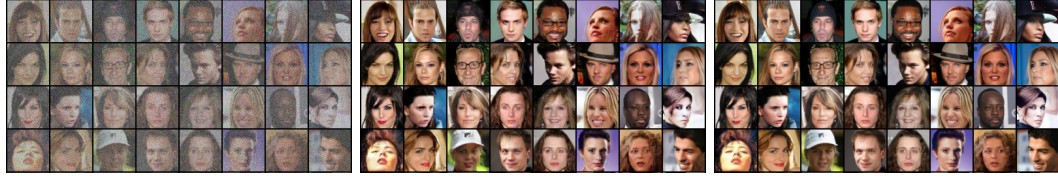

(a) Noisy samples.          (b) Pushforward samples.          (c) Original samples.

Figure 7: OTM for image denoising on test C part of CelebA, $64 \times 64$.

**Colorization.** To create grayscale images, we average the RGB values of each pixel. Figure 8 illustrates image colorization using OTM on the test part of the dataset.

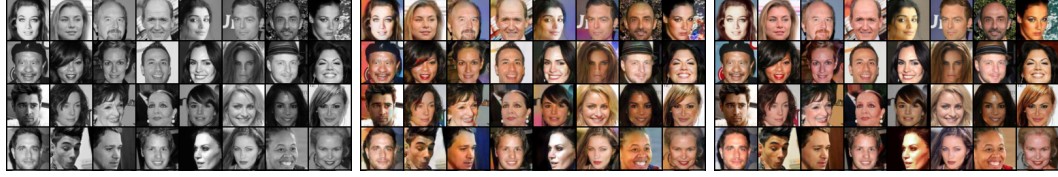

(a) Grayscale samples.          (b) Pushforward samples.          (c) Original samples.

Figure 8: OTM for image colorization on test C part of CelebA, $64 \times 64$.

**Inpainting**. To create incomplete images, we replace the right half of each clean image with zeros. Figure 9 illustrates image inpainting using OTM on the test part of the dataset.

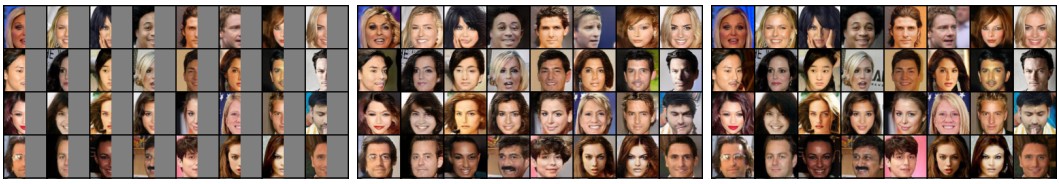

(a) Occluded samples.          (b) Pushforward samples.          (c) Original samples.

Figure 9: OTM for image inpainting on test C part of CelebA, $64 \times 64$.

## 6 CONCLUSION

Our method fits OT maps for the embedded quadratic transport cost between probability distributions. Unlike predecessors, it scales well to high dimensions producing applications of OT maps directly in ambient spaces, such as spaces of images. The performance is comparable to other existing generative models while the complexity of training is similar to that of popular WGANs.

**Limitations.** For distributions $\mu, \nu$ we assume the existence of the OT map between them. In practice, this might not hold for all real-world $\mu, \nu$. Working with equal dimensions, we focus on the quadratic ground cost $\frac{1}{2}\|x - y\|^2$. Nevertheless, our approach extends to other costs $c(\cdot, \cdot)$, see Fan et al. (2021). When the dimensions are unequal, we restrict our analysis to *embedded* quadratic cost $\frac{1}{2}\|Q(x) - y\|^2$ where $Q$ equalizes dimensions. Choosing the embedding $Q$ might not be straightforward in some practical problems, but our evaluation (§5.1) shows that even naive choices of $Q$ work well.

**Potential impact and ethics.** Real-world image restoration problems often do not have paired datasets limiting the application of supervised techniques. In these practical unpaired learning problems, we expect our optimal transport approach to improve the performance of the existing models. However, biases in data might lead to biases in the pushforward samples. This should be taken into account when using our method in practical problems.

**Reproducibility.** The `PyTorch` source code is provided at

    https://github.com/LituRout/OptimalTransportModeling

The instructions to use the code are included in the `README.md` file.

## 7 ACKNOWLEDGMENT

This research was supported by the computational resources provided by Space Applications Centre (SAC), ISRO. The first author acknowledges the funding by HRD Grant No. 0303T50FM703/SAC/ISRO. Skoltech RAIC center was supported by the RF Government (subsidy agreement 000000D730321P5Q0002, Grant No. 70-2021-00145 02.11.2021).

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

# A PROOFS

## A.1 PROOF OF EQUIVALENCE: EQUATION (10) AND (11)

*Proof.* Pick any $T : \mathcal{X} \to \mathcal{Y}$. For every point $x \in \mathcal{X}$ by the definition of the supremum we have

$$\langle x, T(x) \rangle - \psi\left(T(x)\right) \leq \sup_{y \in \mathcal{Y}} \left\{ \langle x, y \rangle - \psi(y) \right\}.$$

Integrating the expression w.r.t. $x \sim \mu$ yields

$$\int_{\mathcal{X}} \{ \langle x, T(x) \rangle - \psi\left(T(x)\right) \} d\mu(x) \leq \int_{\mathcal{X}} \sup_{y \in \mathcal{Y}} \left\{ \langle x, y \rangle - \psi(y) \right\} d\mu(x) = \mathcal{L}_1.$$

Since the inequality holds for all $T : \mathcal{X} \to \mathcal{Y}$, we conclude that

$$\mathcal{L}_2 = \sup_{T : \mathcal{X} \to \mathcal{Y}} \int_{\mathcal{X}} \{ \langle x, T(x) \rangle - \psi\left(T(x)\right) \} d\mu(x) \leq \int_{\mathcal{X}} \sup_{y \in \mathcal{Y}} \left\{ \langle x, y \rangle - \psi(y) \right\} d\mu(x) = \mathcal{L}_1,$$

i.e. $\mathcal{L}_2 \leq \mathcal{L}_1$. Now let us prove that the sup on the left side actually equals $\mathcal{L}_1$. To do this, we need to show that for every $\epsilon > 0$ there exists $T^\epsilon : \mathcal{X} \to \mathcal{Y}$ satisfying

$$\int_{\mathcal{X}} \{ \langle x, T^\epsilon(x) \rangle - \psi\left(T^\epsilon(x)\right) \} d\mu(x) \geq \mathcal{L}_1 - \epsilon.$$

First note that for every $x \in \mathcal{X}$ by the definition of the supremum there exists $y^\epsilon = y^\epsilon(x)$ which provides

$$\langle x, y^\epsilon(x) \rangle - \psi\left(y^\epsilon(x)\right) \geq \sup_{y \in \mathcal{Y}} \left\{ \langle x, y \rangle - \psi(y) \right\} - \epsilon.$$

We take $T^\epsilon(x) = y^\epsilon(x)$ for all $x \in \mathcal{X}$ and integrate the previous inequality w.r.t. $x \sim \mu$. We obtain

$$\int_{\mathcal{X}} \{ \langle x, T^\epsilon(x) \rangle - \psi\left(T^\epsilon(x)\right) \} d\mu(x) \geq \int_{\mathcal{X}} \sup_{y \in \mathcal{Y}} \left\{ \langle x, y \rangle - \psi(y) \right\} d\mu(x) - \epsilon = \mathcal{L}_1 - \epsilon,$$

which is the desired inequality. $\qquad\square$

## A.2 PROOF OF LEMMA 4.1

*Proof.* It is enough to prove that $\overline{\psi^*}(x) = \langle T^*(x), x \rangle - \psi^*\left(T^*(x)\right)$ holds $\mu$-almost everywhere, i.e., $T^*(x) \in \arg\sup_{y \in \mathbb{R}^D} \left\{ \langle x, y \rangle - \psi^*(y) \right\}$. Since $\nu = T^*_\# \mu$, we use (9) with $\psi \leftarrow \psi^*$ to derive

$$\mathcal{W}_2^2(\mu, \nu) - \int_{\mathcal{X}} \frac{1}{2} \|x\|^2 d\mu(x) - \int_{\mathcal{Y}} \frac{1}{2} \|y\|^2 d\nu(y) =$$

$$- \int_{\mathcal{X}} \overline{\psi^*}(x) d\mu(x) - \int_{\mathcal{Y}} \psi^*(y) d\nu(y) = - \int_{\mathcal{X}} \overline{\psi^*}(x) d\mu(x) - \int_{\mathcal{Y}} \psi^*\left(T^*(x)\right) d\mu(x) =$$

$$- \int_{\mathcal{X}} \big[ \underbrace{\overline{\psi^*}(x) + \psi^*\left(T^*(x)\right)}_{\geq \langle T^*(x), x \rangle} \big] d\mu(x) \leq - \int_{\mathcal{X}} \langle T^*(x), x \rangle d\mu(x) = \qquad (17)$$

$$\int_{\mathcal{X}} \frac{1}{2} \|x - T^*(x)\|^2 d\mu(x) - \int_{\mathcal{X}} \frac{1}{2} \|x\|^2 d\mu(x) - \int_{\mathcal{X}} \frac{1}{2} \|T^*(x)\|^2 d\mu(x) =$$

$$\mathcal{W}_2^2(\mu, \nu) - \int_{\mathcal{X}} \frac{1}{2} \|x\|^2 d\mu(x) - \int_{\mathcal{Y}} \frac{1}{2} \|y\|^2 d\nu(y).$$

As a result, inequality (17) becomes the equality, in particular, $\overline{\psi^*}(x) + \psi^*\left(T^*(x)\right) = \langle T^*(x), x \rangle$ holds $\mu$-almost everywhere.

$\qquad\square$

## A.3   PROOF OF LEMMA 4.2

*Proof.* Let $QW_2^2$ denote the $Q$-embedded quadratic cost. We use the change of variables formula to derive

$$QW_2^2(\mu, \nu) = \inf_{\pi \in \Pi(\mu, \nu)} \int_{\mathcal{X} \times \mathcal{Y}} \frac{1}{2} \|Q(x) - y\|^2 d\pi(x, y) =$$

$$\inf_{\pi' \in \Pi(Q_\#\mu, \nu)} \int_{\mathcal{X} \times \mathcal{Y}} \frac{1}{2} \|x - y\|^2 d\pi'(x, y) = W_2^2(Q_\#\mu, \nu), \tag{18}$$

i.e., computing the OT plan for $QW_2^2(\mu, \nu)$ boils down to computing the OT plan for $W_2^2(Q_\#\mu, \nu)$. It follows that $[\text{id}_{\mathbb{R}^H}, T^*(Q(x))]_\#\mu = [\text{id}_{\mathbb{R}^H}, G^*]_\#\mu$ is an OT plan for $QW_2^2(\mu, \nu)$, and $G^*$ is the OT map. Inclusion (15) now follows from Lemma 4.1. $\qquad \square$

## A.4   PROOF OF THEOREM 4.3

*Proof.* Pick any $G' \in \arg\sup_G \mathcal{L}(\hat{\psi}, G) = \arg\sup_G \int_{\mathcal{X}} \left\{ \langle Q(x), G(x) \rangle - \hat{\psi}(G(x)) \right\} d\mu(x)$ or, equivalently, for all $x \in \mathbb{R}^H$, $G'(x) \in \arg\sup_y \left\{ \langle Q(x), y \rangle - \hat{\psi}(y) \right\}$. Consequently, for all $y \in \mathbb{R}^D$

$$\langle Q(x), G'(x) \rangle - \hat{\psi}(G'(x)) \geq \langle Q(x), y \rangle - \hat{\psi}(y),$$

which after regrouping the terms yields

$$\hat{\psi}(y) \geq \hat{\psi}(G'(x)) + \langle Q(x), y - G'(x) \rangle.$$

This means that $Q(x)$ is contained in the subgradient $\partial\hat{\psi}$ at $G'(x)$ for a convex $\hat{\psi}$. Since $\hat{\psi}$ is $\beta$-strongly convex, for points $G(x), G'(x) \in \mathbb{R}^D$ and $Q(x) \in \partial\hat{\psi}(G'(x))$ we derive

$$\hat{\psi}(G(x)) \geq \hat{\psi}(G'(x)) + \langle Q(x), G(x) - G'(x) \rangle + \frac{\beta}{2} \|G'(x) - G(x)\|^2.$$

Regrouping the terms, this gives

$$\left[ \langle Q(x), G'(x) \rangle - \hat{\psi}(G'(x)) \right] - \left[ \langle Q(x), G(x) \rangle - \hat{\psi}(G(x)) \right] \geq \frac{\beta}{2} \|G'(x) - G(x)\|^2.$$

Integrating w.r.t. $x \sim \mu$ yields

$$\epsilon_1 = \mathcal{L}(\hat{\psi}, G') - \mathcal{L}(\hat{\psi}, G) \geq \beta \int_{\mathcal{X}} \frac{1}{2} \|G'(x) - G(x)\|^2 d\mu(x) = \frac{\beta}{2} \cdot \|G - G'\|_{L^2(\mu)}^2. \tag{19}$$

Let $G^*$ be the OT map from $\mu$ to $\nu$. We use $G_\#^*\mu = \nu$ to derive

$$\mathcal{L}(\hat{\psi}, G') = \int_{\mathcal{X}} \left\{ \langle Q(x), G'(x) \rangle - \hat{\psi}(G'(x)) \right\} d\mu(x) + \int_{\mathcal{Y}} \hat{\psi}(y) d\nu(y) =$$

$$\int_{\mathcal{X}} \left\{ \langle Q(x), G'(x) \rangle - \hat{\psi}(G'(x)) \right\} d\mu(x) + \int_{\mathcal{X}} \hat{\psi}(G^*(x)) d\mu(x) =$$

$$\int_{\mathcal{X}} \left\{ \underbrace{\langle Q(x), G'(x) \rangle - \hat{\psi}(G'(x)) + \hat{\psi}(G^*(x))}_{\geq \langle Q(x), G^*(x) \rangle + \beta \frac{1}{2} \|G' - G^*\|^2} \right\} d\mu(x) \geq$$

$$\int_{\mathcal{X}} \langle Q(x), G^*(x) \rangle d\mu(x) + \beta \int_{\mathcal{X}} \frac{1}{2} \|G' - G^*\|^2 d\mu(x). \tag{20}$$

Let $\psi^*$ be an optimal potential in (14). Thanks to Lemma 4.2, we have

$$\inf_\psi \sup_G \mathcal{L}(\psi, G) = \mathcal{L}(\psi^*, G^*) =$$

$$\int_{\mathcal{X}} \left\{ \langle Q(x), G^*(x) \rangle - \psi^*(G^*(x)) \right\} d\mu(x) + \int_{\mathcal{Y}} \psi^*(y) d\nu(y) =$$

$$\int_{\mathcal{X}} \left\{ \langle Q(x), G^*(x) \rangle - \psi^* \big(G^*(x)\big) \right\} d\mu(x) + \int_{\mathcal{X}} \psi^* \big(G^*(x)\big) d\mu(x) =$$
$$\int_{\mathcal{X}} \langle Q(x), G^*(x) \rangle d\mu(x) \qquad (21)$$

By combining (20) with (21), we obtain

$$\epsilon_2 = \mathcal{L}(\hat{\psi}, G') - \mathcal{L}(\psi^*, G^*) \geq \beta \int_{\mathcal{X}} \frac{1}{2} \|G' - G^*\|^2 d\mu(x) = \frac{\beta}{2} \cdot \|G' - G^*\|_{L^2(\mu)}^2 \qquad (22)$$

The right-hand inequality of (16) follows from the triangle inequality combined with (19) and (22). The middle inequality of (16) follows from (Korotin et al., 2021a, Lemma A.2) and $G^*_{\#}\mu = \nu$.

Now we prove the left-hand inequality of (16). Let $\mathcal{I}$ be the feature extractor of the pre-trained InceptionV3 neural networks. FID score between generated (fake) $\hat{G}_{\#}\mu$ and data distribution $\nu$ is

$$\text{FID}(\hat{G}_{\#}\mu, \nu) = \text{FD}(\mathcal{I}_{\#}\hat{G}_{\#}\mu, \mathcal{I}_{\#}\nu) \leq 2 \cdot \mathcal{W}_2^2(\mathcal{I}_{\#}\hat{G}_{\#}\mu, \mathcal{I}_{\#}\nu), \qquad (23)$$

where $\text{FD}(\cdot, \cdot)$ is the Fréchet distance which lower bounds $2 \cdot \mathcal{W}_2^2$, see (Dowson & Landau, 1982). Finally, from (Korotin et al., 2021a, Lemma A.1) it follows that

$$\mathcal{W}_2^2(\mathcal{I}_{\#}\hat{G}_{\#}\mu, \mathcal{I}_{\#}\nu) \leq L^2 \cdot \mathcal{W}_2^2(\hat{G}_{\#}\mu, \nu). \qquad (24)$$

Here $L$ is the Lipschitz constant of $\mathcal{I}$. We combine (23) and (24) to get the left-hand inequality in (16). $\qquad \square$

## B EXPERIMENTAL DETAILS

We use the PyTorch framework. All the experiments are conducted on $2 \times$V100 GPUs. We compute inception and FID scores with the official implementation from OpenAI[1] and TTUR[2]. The compared results are taken from the respective papers or publicly available source codes.

### B.1 GENERAL TRAINING DETAILS

**MNIST (LeCun et al., 1998).** On MNIST, we use $x \in \mathbb{R}^{192}$ and $y \in \mathbb{R}^{32 \times 32}$. The batch size is 64, learning rate $2 \cdot 10^{-4}$, optimizer Adam (Kingma & Ba, 2014) with betas $(0, 0.9)$, gradient optimality coefficient $\lambda = 10$, and the number of training epochs $T = 30$. We observe stable training while updating $\psi$ once in multiple $G$ updates, i.e., $k_G = 2$ and $k_{\psi} = 1$.

**CIFAR10 (Krizhevsky et al., 2009).** We use all 50000 samples while training. The latent vector $x \in \mathbb{R}^{192}$ and $y \in \mathbb{R}^{32 \times 32 \times 3}$, batch size 64, $\lambda = 10$, $k_G = 1$, $k_{\psi} = 1$, $T = 1000$, Adam optimizer with betas $(0, 0.9)$, and learning rate $2 \cdot 10^{-4}$ for $G$ and $1 \cdot 10^{-3}$ for $\psi$.

**CelebA (Liu et al., 2015).** We use $x \in \mathbb{R}^{192}$ and $y \in \mathbb{R}^{64 \times 64 \times 3}$. The images are first cropped at the center with size 140 and then resized to $64 \times 64$. We consider all 202599 samples. We use Adam with betas $(0, 0.9)$, $T = 200$, $K_G = 2$, $K_{\psi} = 1$ and learning rate $2 \cdot 10^{-4}$.

**Image restoration.** In the unpaired image restoration experiments, we use Adam optimizer with betas $(0, 0.9)$, $K_G = 5$, $K_{\psi} = 1$, $\lambda = 0$, learning rate $1 \cdot 10^{-4}$ and train for $T = 300$ epochs.

**CelebA128x128 (Liu et al., 2015).** On this dataset, we resize the cropped images as in **CelebA** to $128 \times 128$, i.e. $y \in \mathbb{R}^{128 \times 128 \times 3}$. Here, $K_G = 5$, $K_{\psi} = 1$, $\lambda = 0.01$, learning rate $1 \cdot 10^{-4}$ and betas=$(0.5, 0.999)$. The batch size is reduced to 16 so as to fit in the GPU memory.

**Anime128x128[3].** This dataset consists of 500000 high resolution images. We resize the cropped images as in **CelebA** to $128 \times 128$, i.e. $y \in \mathbb{R}^{128 \times 128 \times 3}$. Here, $K_G = 5$, $K_{\psi} = 1$, $\lambda = 0.01$, learning rate $2 \cdot 10^{-4}$, batch size 16, and betas=$(0, 0.9)$.

**Toy datasets.** The dimension is $D = H = 2$, total number of samples is 10000. We use the batch size 400, $\lambda = 0.1$, $K_{\psi} = 1$, $K_G = 16$, and

---

[1]IS: https://github.com/openai/improved-gan/tree/master/inception_score

[2]FID: https://github.com/bioinf-jku/TTUR

[3]Anime: https://www.kaggle.com/reitanaka/alignedanimefaces

$T = 100$. The optimizer is Adam with betas $(0.5, 0.99)$ and learning rate $1 \cdot 10^{-3}$. We use the following datasets: Gaussian to mixture of Gaussians[4], two moons (`sklearn.datasets.make_moons`), circles (`sklearn.datasets.make_circles`), gaussian to S-curve (`sklearn.datasets.make_s_curve`), and gaussian to swiss roll (`sklearn.datasets.make_swiss_roll`).

**Wasserstein-2 benchmark (Appendix B.2).** The dimension is $D = H = 64 \times 64 \times 3$. We use batch size 64, $\lambda = 0$, $K_\psi = 1$, $K_G = 5$, learning rate $10^{-4}$, and Adam optimizer with default betas.

## B.2 EVALUATION ON THE CONTINUOUS WASSERSTEIN-2 BENCHMARK

To empirically show that the method recovers the optimal transport maps well on equal dimensions, we evaluate it on the recent continuous Wasserstein-2 benchmark by Korotin et al. (2021b). The benchmark provides a number of artificial test pairs $(\mu, \nu)$ of continuous probability distributions with analytically known OT map $T^*$ between them.

For evaluation, we use the "Early" images benchmark pair ($D = 12288$), see (Korotin et al., 2021b, §4.1) for details. We adopt the $\mathcal{L}^2$-unexplained percentage metric (Korotin et al., 2021a, §5.1) to quantify the recovered OT map $\hat{T}$: $\mathcal{L}^2\text{-UVP}(\hat{T}) = 100 \cdot \|\hat{T} - T^*\|^2 / \text{Var}(\nu)\%$. For our method the $\mathcal{L}_2$-UVP metric is only $\approx 1\%$, see Table 4. This is comparable to the best $\lceil$MM:R$\rceil$ method which the authors evaluate on their benchmark. The qualitative results are given in Figure 10.

| Method | $\mathcal{L}^2$-UVP↓ |
|---|---|
| $\lceil$MM:R$\rceil$ | 1.4% |
| OTM (ours) | 1.32% |

Table 4: $\mathcal{L}^2$-UVP metric of the recovered transport map on the "Early" images benchmark pair.

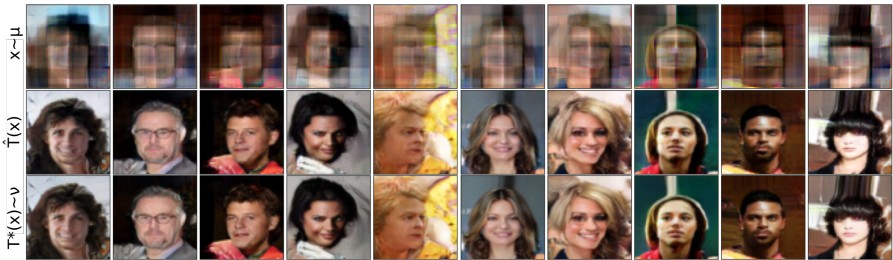

Figure 10: Qualitative results of OTM on the "Early" images benchmark pair $(\mu, \nu)$ by Korotin et al. (2021b). The 1st line shows samples $x \sim \mu$, the 2nd line shows fitted OT map $\hat{T}(x)$, and the 3rd line shows the corresponding optimal map $T^*(x) \sim \nu$.

## B.3 FURTHER DISCUSSION AND EVALUATION

**Generative modeling.** In the experiments, we use the gradient penalty on $\psi$ for better stability of optimization. The penalty is intended to make the gradient norm of the optimal WGAN critic equal to 1 (Gulrajani et al., 2017, Corollary 1). This condition does not necessarily hold for optimal $\psi^*$ in our case and consequently might introduce bias to optimization.

To address this issue, we additionally tested an alternative regularization which we call the *gradient optimality*. For every optimal potential $\psi^*$ and map $G^*$ of problem (14), we get from Lemma 4.2:

$$\nabla_G \left\{ \mathbb{E}_{x \sim \mu} \left[ \langle Q(x), G^*(x) \rangle - \psi^* (G^*(x)) \right] \right\} = \mathbb{E}_{x \sim \mu} [Q(x)] - \mathbb{E}_{x \sim \mu} [\nabla \psi^* (G^*(x))] = 0. \quad (25)$$

Since $\mu$ is normal noise distribution and $Q(x)$ is naive upscaling (§5.1), the above expression simplifies to $\mathbb{E}_{x \sim \mu} \nabla \psi^* (G^*(x)) = 0$. Based on this property, we establish the following regularizer $\lambda \| \mathbb{E}_{x \sim \mu} \nabla \psi (G(x)) \|$ for $\lambda > 0$ and add this to $\mathcal{L}_\psi$ in our Algorithm 1.

While gradient penalty considers expectation of norm, gradient optimality considers norm of expectation. The gradient optimality is always non-negative and vanishes at the optimal point.

---

[4]`https://github.com/AmirTag/OT-ICNN`

We conduct additional experiments with the gradient optimality and compare FID scores for different $\lambda$ in Table 5. It leads to an improvement of FID score from the earlier 7.7 with the gradient penalty to the current 6.5 with the gradient optimality on CelebA (Table 2).

**Unpaired restoration.** In the unpaired restoration experiments (§5.2), we test OTM with the gradient penalty to make a fair comparison with the baseline WGAN-GP. We find OTM without regularization, i.e., $\lambda = 0$ works better than OTM-GP (Table 3). In practice, more $G$ updates for a single $\psi$ update works fairly well (§B.1).

| $\lambda$ | FID$\downarrow$ |
|---|---|
| 0.001 | 16.91 |
| 0.01 | 16.22 |
| 0.1 | 16.70 |
| 1.0 | 10.01 |
| 10 | 6.50 |

Table 5: Ablation study of gradient optimality in OTM.

### B.4 ADDITIONAL QUALITATIVE RESULTS

OTM works with both the grayscale and color embeddings of noise in the ambient space.

**CelebA128x128.** Figure 11 shows the grayscale embedding $Q(x)$, the recovered transport map $\hat{G}(x)$, and independently drawn real samples $y \sim \nu$.

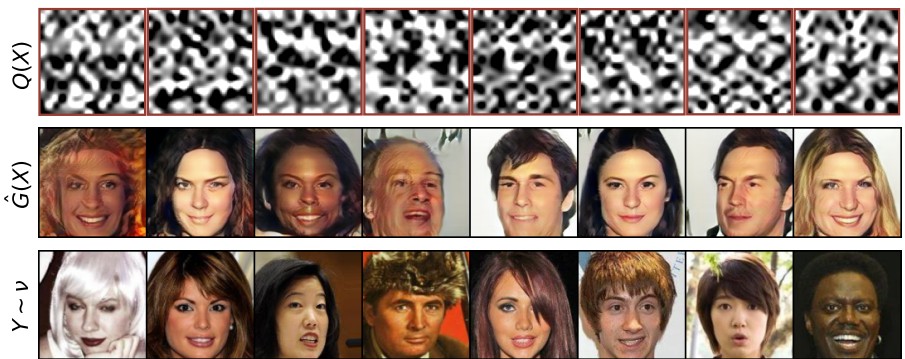

Figure 11: OTM between 128-dimensional noise and CelebA, $128 \times 128$. The 1st line shows the **grayscale** embedding $Q$ (repeating bicubic upscaling of a noise, $16 \times 8$), the 2nd line shows corresponding generated samples, and the 3rd line shows random samples from the dataset.

**Anime128x128.** Figure 12 shows the color embedding $Q(x)$, the recovered transport map $\hat{G}(x)$, and independently drawn real samples $y \sim \nu$.

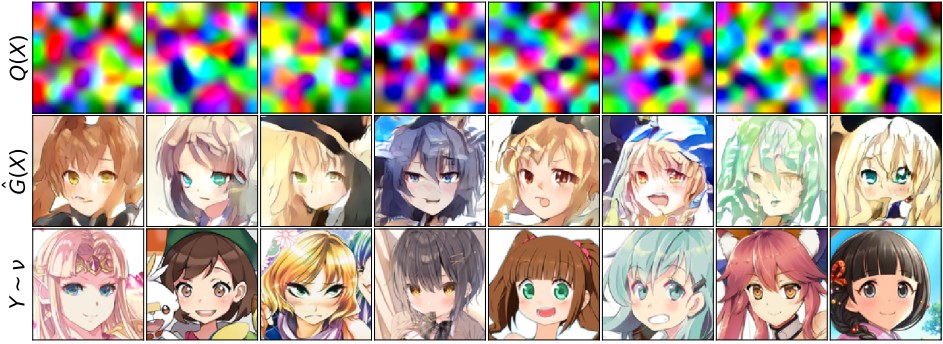

Figure 12: OTM between 192-dimensional noise and Anime, $128 \times 128$. The 1st line shows the **color** embedding $Q$ (bicubic upscaling of a noise, $3 \times 8 \times 8$), the 2nd line shows corresponding generated samples, and the 3rd line shows random samples from the dataset.

The extended qualitative results with color embedding on **MNIST**, **CIFAR10**, and **CelebA** are shown in Figure 13a, Figure 13b, and Figure 13c respectively. Table 6 shows quantiative results on MNIST. The color embedding $Q$ is bicubic upscaling of a noise in $\mathbb{R}^{3 \times 8 \times 8}$. The samples are

Table 6: Results on MNIST dataset.

| Model | Related Work | FID ↓ |
|-------|-------------|-------|
| VAE | Kingma & Welling (2013) | 23.8±0.6 |
| LSGAN | Mao et al. (2017) | 7.8±0.6 |
| BEGAN | Berthelot et al. (2017) | 13.1±1.0 |
| WGAN | Arjovsky et al. (2017) | 6.7±0.4 |
| SIG | Dai & Seljak (2021) | 4.5 |
| AE-OT | An et al. (2020a) | 6.2±0.2 |
| AE-OT-GAN | An et al. (2020b) | 3.2 |
| OTM | Ours | 2.4 |

generated randomly (uncurated) by fitted optimal transport maps between noise and ambient space, e.g., spaces of high-dimensional images. Figure 14 illustrates latent space interpolation between the generated samples. Figure 15 shows denoising of images with varying levels of $\sigma = 0.1, 0.2, 0.3, 0.4$ by the model trained with $\sigma = 0.3$.

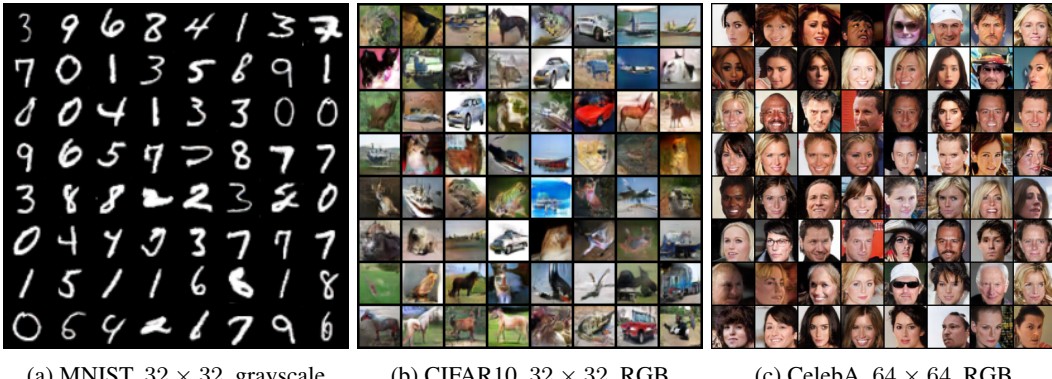

(a) MNIST, $32 \times 32$, grayscale  (b) CIFAR10, $32 \times 32$, RGB  (c) CelebA, $64 \times 64$, RGB

Figure 13: Randomly generated MNIST, CIFAR10, and CelebA samples by our method (OTM).

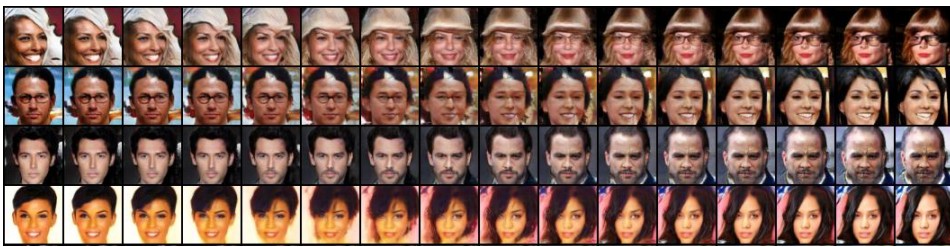

Figure 14: OTM for latent space interpolation on CelebA, $64 \times 64$. Extended samples.

**Toy datasets.** Figure 16 shows the results of our method and related approaches (§3) on a toy 2D dataset. Figure 17 shows the results of our method applied to other toy datasets.

## B.5 NEURAL NETWORK ARCHITECTURES

This section contains architectures on **CIFAR10** (Table 7), **CelebA** (Table 8), **CelebA** $128 \times 128$ (Figure 11) generation, **image restoration** tasks (Table 9), evaluation on the toy 2D datasets and **Wasserstein-2 images benchmark** (Table 4).

In the unpaired restoration tasks (§5.2), we use UNet architecture for transport map $G$ and convolutional architecture for potential $\psi$. Similarly to (Song & Ermon, 2019), we use BatchNormaliation (BN) and InstanceNormalization+ (INorm+) layers. In the ResNet architectures, we use the ResidualBlock of NCSN (Song & Ermon, 2019).

In the toy 2D examples, we use a simple multi-layer perceptron with 3 hidden layers consisting of 128 neurons each and LeakyReLU activation. The final layer is linear without any activation.

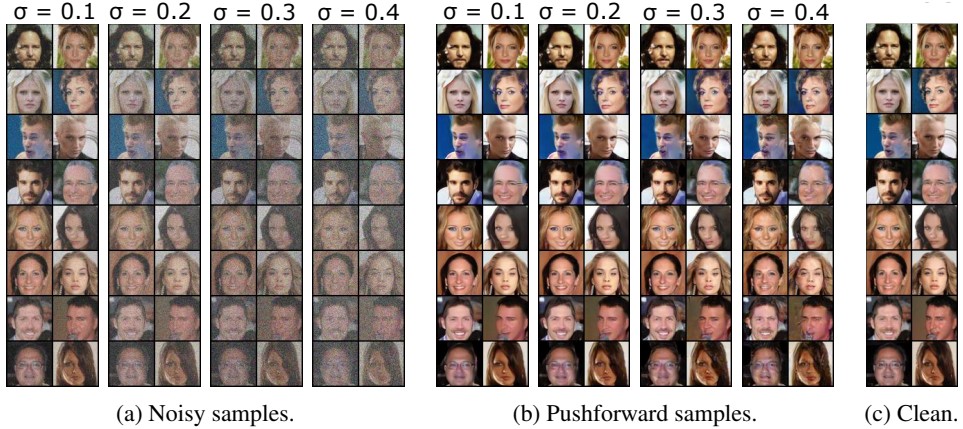

(a) Noisy samples.  (b) Pushforward samples.  (c) Clean.

Figure 15: OTM for image denoising for varying levels of noise on test C part of CelebA, $64 \times 64$.

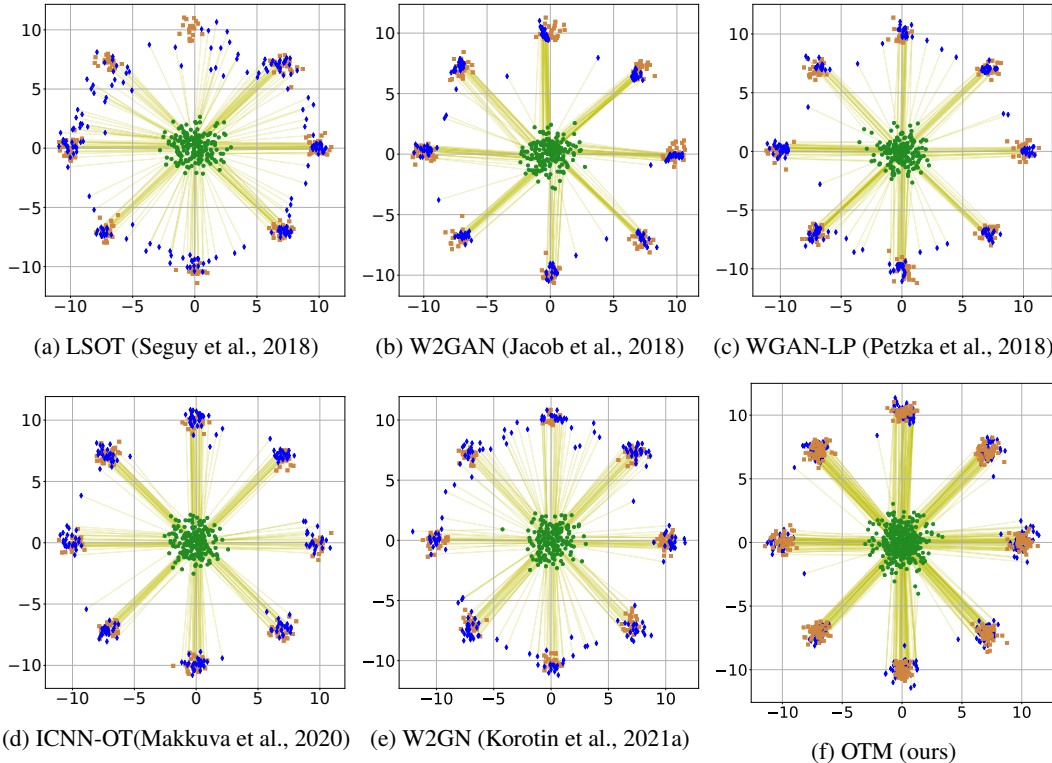

(a) LSOT (Seguy et al., 2018)  (b) W2GAN (Jacob et al., 2018)  (c) WGAN-LP (Petzka et al., 2018)

(d) ICNN-OT(Makkuva et al., 2020)  (e) W2GN (Korotin et al., 2021a)  (f) OTM (ours)

Figure 16: Mapping between a Gaussian and a Mixture of 8 Gaussians in 2D by various methods. The colors green, blue, and peru represent input, pushforward, and output samples respectively.

The transport map $G$ and potential $\psi$ architectures on MNIST $32 \times 32$, CelebA $128 \times 128$, and Anime $128 \times 128$ are the generator and discriminator architectures of WGAN-QC Liu et al. (2019) respectively.

In the evaluation on the Wasserstein-2 benchmark, we use publicly available Unet[5] architecture for transport map $T$ and WGAN-QC discriminator's architecture for $\psi$ (Liu et al., 2019). These neural network architectures are the same as the authors of the benchmark use.

---

[5]https://github.com/milesial/Pytorch-UNet

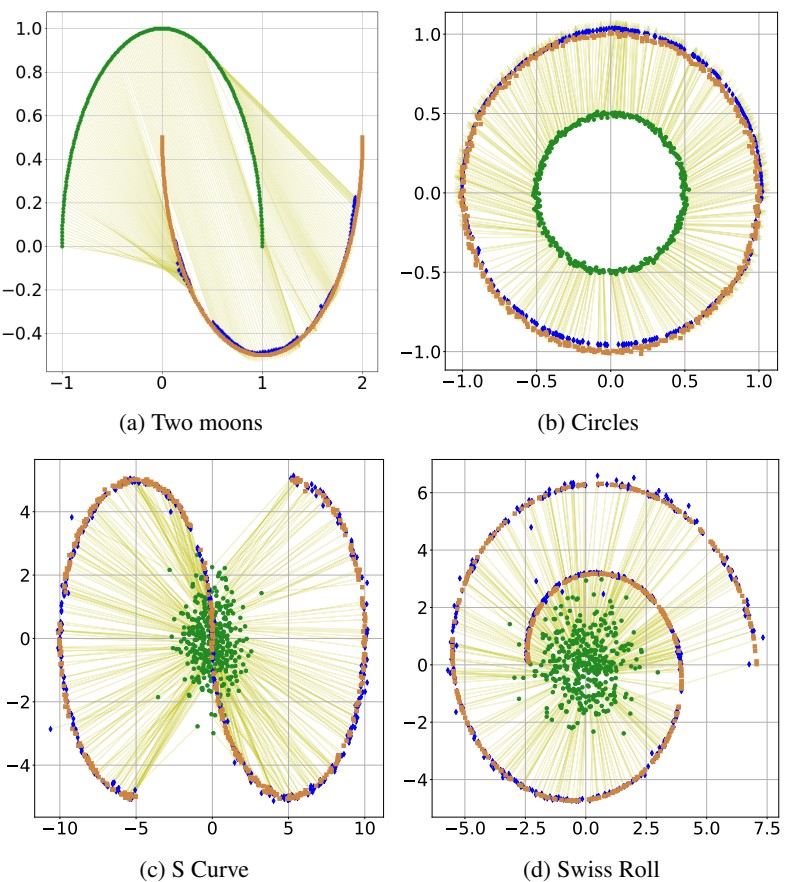

(a) Two moons

(b) Circles

(c) S Curve

(d) Swiss Roll

Figure 17: OTM on toy datasets, $D = 2$. Here, the colors green, blue, and peru represent input, pushforward, and output samples respectively.

Table 7: Architectures for generation task on CIFAR10, $32 \times 32$.

| $G(.)$ |
|---|
| Noise: $x \in \mathbb{R}^{128}$ |
| Linear, Reshape, output shape: $[128 \times 4 \times 4]$ |
| ResidualBlock Up, output shape: $[128 \times 8 \times 8]$ |
| ResidualBlock Up, output shape: $[128 \times 16 \times 16]$ |
| ResidualBlock Up, output shape: $[128 \times 32 \times 32]$ |
| Conv, Tanh, output shape: $[3 \times 32 \times 32]$ |

| $\psi(.)$ |
|---|
| Target: $y \in \mathbb{R}^{3 \times 32 \times 32}$ |
| ResidualBlock Down, output shape: $[128 \times 16 \times 16]$ |
| ResidualBlock Down, output shape: $[128 \times 8 \times 8]$ |
| ResidualBlock, output shape: $[128 \times 8 \times 8]$ |
| ResidualBlock, output shape: $[128 \times 8 \times 8]$ |
| ReLU, Global sum pooling, output shape: $[128 \times 1 \times 1]$ |
| Linear, output shape: $[1]$ |

Table 8: Architectures for generation task on Celeba, $64 \times 64$.

| $G(.)$ |
| --- |
| Noise: $x \in \mathbb{R}^{128}$ |
| ConvTranspose, BN, LeakyReLU, output shape: $[256 \times 1 \times 1]$ |
| ConvTranspose, BN, LeakyReLU, output shape: $[512 \times 4 \times 4]$ |
| Conv, PixelShuffle, BN, LeakyReLU, output shape: $[512 \times 8 \times 8]$ |
| Conv, PixelShuffle, BN, LeakyReLU, output shape: $[512 \times 16 \times 16]$ |
| Conv, PixelShuffle, BN, LeakyReLU, output shape: $[512 \times 32 \times 32]$ |
| ConvTranspose, Tanh, output shape: $[3 \times 64 \times 64]$ |

| $\psi(.)$ |
| --- |
| Target: $y \in \mathbb{R}^{3 \times 64 \times 64}$ |
| Conv, output shape: $[128 \times 64 \times 64]$ |
| ResidualBlock Down, output shape: $[256 \times 32 \times 32]$ |
| ResidualBlock Down, output shape: $[256 \times 16 \times 16]$ |
| ResidualBlock Down, output shape: $[256 \times 8 \times 8]$ |
| ResidualBlock Down, output shape: $[128 \times 4 \times 4]$ |
| Conv, output shape: $[1]$ |

Table 9: Architectures for restoration tasks on CelebA, $64 \times 64$.

| $G(.)$ |
| --- |
| Input: $x \in \mathbb{R}^{3 \times 64 \times 64}$ |
| Conv, BN, LeakyReLU, output shape: $[256 \times 64 \times 64]$ |
| Conv, LeakyReLU, AvgPool, output shape: $[256 \times 32 \times 32]$, x1 |
| Conv, LeakyReLU, AvgPool, output shape: $[256 \times 16 \times 16]$, x2 |
| Conv, LeakyReLU, AvgPool, output shape: $[256 \times 8 \times 8]$, x3 |
| Conv, LeakyReLU, AvgPool, output shape: $[256 \times 4 \times 4]$, x4 |
| Nearest Neighbour Upsample, Conv, BN, ReLU, output shape: $[256 \times 8 \times 8]$, y3 |
| Add (y3, x3), output shape: $[256 \times 8 \times 8]$, y3 |
| Nearest Neighbour Upsample, Conv, BN, ReLU, output shape: $[256 \times 16 \times 16]$, y2 |
| Add (y2, x2), output shape: $[256 \times 16 \times 16]$, y2 |
| Nearest Neighbour Upsample, Conv, BN, ReLU, output shape: $[256 \times 32 \times 32]$, y1 |
| Add (y1, x1), output shape: $[256 \times 32 \times 32]$, y1 |
| Nearest Neighbour Upsample, Conv, BN, ReLU, output shape: $[256 \times 64 \times 64]$, y |
| Add (y, x), output shape: $[256 \times 64 \times 64]$, y |
| ConvTranspose, Tanh, output shape: $[3 \times 64 \times 64]$ |

| $\psi(.)$ |
| --- |
| Target: $y \in \mathbb{R}^{3 \times 64 \times 64}$ |
| Conv, LeakyReLU, AvgPool, output shape: $[256 \times 32 \times 32]$ |
| Conv, LeakyReLU, AvgPool, output shape: $[256 \times 16 \times 16]$ |
| Conv, LeakyReLU, AvgPool, output shape: $[256 \times 8 \times 8]$ |
| Conv, LeakyReLU, AvgPool, output shape: $[256 \times 4 \times 4]$ |
| Linear, output shape: $[1]$ |

