# OpenReview forum: "Generative Modeling with Optimal Transport Maps"
_ICLR.cc/2022/Conference — ICLR 2022 Poster_

### Official Review · Reviewer_4T3P · 2021-11-02

**Correctness:** 3
**Technical Novelty And Significance:** 3
**Empirical Novelty And Significance:** 3
**Recommendation:** 6
**Confidence:** 5

**Main Review:**

This paper has some strengths. 1) the proposed method can apply OT maps directly in ambient space, and fit OT maps for Q-embedded quadratic cost between located on the spaces of unequal and equal dimensions. 2) The boundary error of OT maps is analyzed theoretically.
This paper has some weaknesses. 1) Compared with OT maps as loss function and OT maps as generation model, what is computational complexity of this paper. 2) what is the detailed model diagram of this paper, and how do you choose appropriate Q-embedding, what exactly is Q-embedding ?


**Summary Of The Paper:**

This paper proposes a min-max optimization algorithm to apply OT maps directly in ambient space. And that extends the method the case when the input and output distributions are located in the spaces of different dimensions and derive error bounds for the computed OT map.
Contributions of this paper have three aspects: 1) the end-to-end algorithm is given to fit OT maps for Q-embedded quadratic cost, i.e., if two distributions located on the spaces of equal dimensions, the identity embedding Q(x)≡x (Wasserstein-2 distance); if not, choosing Q-embedded quadratic cost. 2) The theoretical analysis the error bounds is proved. 3) this paper demonstrates large-scale applications of OT maps in CV tasks, image generation and image restoration.


**Summary Of The Review:**

The idea of this paper is very clear, the theoretical analysis is rigorous, and the experimental results can fully illustrate the performance and effectiveness of the method. But there are a few problems, The c-transform $u^{c}(x)$ on page3 should be $u^{c}(y)$, The computational complexity of the algorithm is not reflected in the text, what is Q-embedding? what is the relationship between it and the generated mapping?

---

> ### Author Response · Authors · 2021-11-15
> **Answer to Reviewer 4T3P**
>
> Dear reviewer 4T3P,
>
> Thanks for your thoughtful review. We appreciate that you are positive about our theoretical analysis and the experimental results. Please find below the answers to your comments and questions.
>
> **(1) Complexity of the Algorithm**
>
> The time and memory complexity of our algorithm are comparable to the models using **OT cost as the loss**, e.g., WGAN. We detail this in Section 4.3 after algorithm 1. As reviewer fFFr noted, the structure of our model is very similar to that of WGAN, which is a strength in terms of computational complexity.
>
>
> The update of the potential $\psi_{\omega}$ in our algorithm is the same as the update of the discriminator in WGAN. The update of generator $G_{\theta}$ requires computing an additional term $-\langle Q(\cdot),G_{\theta}(\cdot)\rangle$, where $Q$ is an embedding. In the unpaired image restoration experiments (section 5.2), $Q$ is the identity map; in the generative modeling (section 5.1), $Q$ is the bicubic upscaling. These functions are easy to compute compared to $G_{\theta}(\cdot)$ and $\psi_{\omega}(\cdot)$ which are neural networks. Therefore, computing our generator's loss $\psi_{\omega}(G_{\theta}(\cdot))-\langle Q(\cdot),G_{\theta}(\cdot)\rangle$ is only up to a small constant factor harder than the generator loss $\psi_{\omega}(G_{\theta}(\cdot))$ in WGAN.
>
> Comparing the complexity with the methods that use **OT map as the generative model** is tricky. As we note in Section 3, existing OT map methods are mostly applied in latent spaces of autoencoders. To achieve sufficient generative performance, a precise perceptual/adversarial autoencoder is required. The training complexity of such an autoencoder is itself very high and is comparable to GANs/WGANs.
>
> **(2) Diagram of the method; the $Q$-embedding.**
>
> Following your suggestion, we included the detailed diagram of our method, see Figure 4 of Section 4.2.
>
> **(3) How to choose $Q$-embedding?**
>
> Our method searches for the measure-preserving ($G*_{\#}\mu=\nu$) map $G^*$ which minimizes $\frac{1}{2}\|Q(x)-G(x)\|^{2}$, i.e. is the closest map to the embedding $Q$ in the $\mathcal{L}^{2}(\mu)$ sense. Thus, the embedding $Q$ can be viewed as the prior preference.
>
> In the image restoration, $Q(x)=x$ is a reasonable choice since we want the output $G(x)$ not to significantly differ from the input $x$. In image generation, we do not care about the structure of the map $G$, so we choose naive straightforward embedding $Q$ -- upscaling of noise, see Figure 4. Probably, advanced choices of $Q$ are useful to improve the structure of the generative model, e.g., disentangle the latent space. Studying this question is a promising avenue for future work.
>
> **Concluding remarks.** Please respond to our post to let us know if the clarifications above suitably address your concerns about our work. We are happy to address any remaining points during the discussion phase; if the responses above are sufficient, we kindly ask that you consider raising your score.

---

> > ### Author Response · Authors · 2021-11-28
> > **Looking forward to your final feedback**
> >
> > Dear reviewer 4T3P,
> >
> > We thank you for your review and appreciate your time reviewing our paper.
> >
> > The end of the discussion phase is approaching. We would be grateful if we could hear your feedback regarding our revision and answers to the reviews. We are happy to address any remaining points during the remaining period.
> >
> > Thanks in advance,
> >
> > the authors of "Generative Modeling with Optimal Transport Maps"

---

### Official Review · Reviewer_fFFr · 2021-11-02

**Correctness:** 4
**Technical Novelty And Significance:** 3
**Empirical Novelty And Significance:** 3
**Recommendation:** 8
**Confidence:** 4

**Main Review:**

This paper sheds new light on using OT maps as generators by exploiting the OT map from latent space to data space with unequal dimensions. This is in contrast with previous approaches that manipulate distributions in the latent space.

The theoretical results of the error analysis of the proposed method are solid and novel in computer vision field. The proposed method does not add a sophisticated mechanism to compute the OT map compared to WGAN. Instead, the structure is very similar to that of WGAN model, which is a strength in terms of computational complexity.

Extensive evaluation of the proposed method with previous methods is thoroughly conducted on both synthetic and real-world datasets. The proposed method performs comparable or better in terms of visual quality of the generated images as well as FID scores than competitive methods.

A minor concern is that, as pointed out by previous studies (e.g., An 2020(a)), the OT map is generally highly discontinuous which makes approximating this map by DNNs tricky. I think this is probably one of the reasons why some previous methods avoid using DNNs to compute OT/generator maps (in contrast with the proposed method). A discussion on the singularities of the generator/OT map would be helpful for a deeper understanding of the proposed method.

**Summary Of The Paper:**

In this paper, a novel optimal transport (OT) map based generative model is proposed. The OT map is computed as the generator map from noise distribution to data distribution. Different from previous works that mainly focus on manipulating distributions in latent space, the proposed method directly computes the OT map in the data/ambient space. One of the benefits of this approach is that the FID is controlled by the W-2 loss in the data space (i.e., theorem 4.3). Experiment results show that the proposed method has comparable or better performance than competitive methods on various tasks.

**Summary Of The Review:**

Overall the paper is well-presented, with novel theoretical and empirical results. I would like to recommend accepting this paper.

---

> ### Author Response · Authors · 2021-11-15
> **Answer to Reviewer fFFr**
>
> Dear reviewer fFFr,
>
> Thanks for your thoughtful review. We appreciate that you are positive about our results and recognize our efforts in scaling optimal transport maps from latent space to data space with unequal dimensions. Please find below the answer to your comment.
>
> **(1) Learning discontinuous optimal maps with deep neural networks (NN)**
>
> We agree that the optimal map $G^*:\mathbb{R}^{H}\rightarrow\mathbb{R}^{D}$ from $\mu$ to $\nu$ might be discontinuous. Despite this, when $\mu$ has a compact support $\mathcal{M}\subset\mathbb{R}^{D}$, there always exists a continuous NN $\hat{G}:\mathbb{R}^{H}\rightarrow\mathbb{R}^{D}$ which approximates $G^*$ in $\mathcal{L}^{2}(\mu)$ norm with arbitrarily small positive error.
>
> **Proof.** The continuous functions $\mathcal{C}(\mathcal{M})$ are dense in $\mathcal{L}^{2}(\mu)$ w.r.t. $\mathcal{L}^{2}(\mu)$ norm (Folland, 1999, Proposition 7.9).  NNs are dense in $\mathcal{C}(\mathcal{M})$ w.r.t. $\mathcal{L}^{\infty}$ norm (Kidger \& Lyons, 2020, Theorem 3.2), and, consequently $\mathcal{L}^{2}(\mu)$ norm. We combine these two observations and conclude that NNs are dense in $\mathcal{L}^{2}(\mu)$ w.r.t. $\mathcal{L}^{2}(\mu)$ norm. Q.E.D.
>
> In practice, we did not experience issues with learning discontinuous maps with deep NNs. We added additional 2D qualitative results (see Figure 18 in Appendix B.3) showing how our method performs when the true transport map is discontinuous.
>
> **Concluding remarks.** Please respond to our post to let us know if the clarifications above suitably address your concerns about our work. We are happy to address any remaining points during the discussion phase.
>
> **References**
>
> Folland, G. B. (1999). Real analysis: modern techniques and their applications (Vol. 40). John Wiley\& Sons.
>
> Kidger, P., \& Lyons, T. (2020, July). Universal approximation with deep narrow networks. In Conference on learning theory (pp. 2306-2327). PMLR.

---

### Official Review · Reviewer_qNeK · 2021-11-03

**Correctness:** 3
**Technical Novelty And Significance:** 3
**Empirical Novelty And Significance:** 2
**Recommendation:** 6
**Confidence:** 3

**Main Review:**

This work addresses an important problem and proposes an interesting solution using the OT map between input and output distributions.
Experimental results show that the method performs similar to other existing methods.

**Paper organization/presentation**

- We need to wait till the middle of Section 4 to understand what the paper proposes. The introduction to the proposed idea is too long (even if  some of the concepts need to be provided,  a lot of that content can probably go to an Appendix).

- Lemma 4.1 seems the relevant one. Maybe this could be better introduced in the paper (there's no  connection to the previous paragraphs, so it's hard to follow the story).

- The paragraph following Lemma 4.1, that tries to connect the Lemma to the previous work is not particularly clear. The ultimate question is: have people tried this idea of estimating an OT map before? and if so, how is the proposed solution different from previous work (technical difference, algorithmic difference,....).

- The technical extension to unequal dimensions seems straightforward if you have Q.

- The paper discusses in multiple places the relation to previous work. This makes it hard to follow (the presentation is a little not linear, one needs to jump from one place to the other to get all the pieces).

**Technical comments**
- Most image restoration problems are ill-posed: there are multiple clean images that could lead to the same low-quality observation. This implies that it won't exist a deterministic OT plan (so no 1-1- map). This is particularly obvious in the case of image colorization: from a gray image estimate a possible RGB image. This raises the question of what is the formulation doing in this case? There seems to be a mismatch between the mathematical formulation and the practical use in inverse problems.

- The paper claims optimality of the restoration map, but in most inverse problems it is unclear wether this mapping exists (in general there's a loss of information so no 1-1 mapping between low-quality - high-quality image).

- One of the claims is that previous work using OT map didn't address the problem in the input space dimension. The paper does not provide intuition or a mathematical justification why the proposed formulation should work better than the other methods when addressing the problem in the original dimension.

**Experiments**
- The paper claims to be able to work on high-dimensional data, but in terms of image size the experiments consist of images of at most 64x64 RGB pixels. It's true that this could represent a high-dimensional dataset, but in general most of the problems with generated/restored images are visible at larger resolutions (this is particularly the case for image restoration).


**Typos**
- pag 3. Duality paragraph. $u^c(x)$ → $u^c(y)$.
- pag 4. Alexandroff theory (?)
- pag 4. Figure 3. This figure is not particularly illustrative.
- pag4 - pag5. Indentation and presentation of equations (8) to (11) is confusing.
- pag 5.  const($\mu$,$\nu$) → cost($\mu$,$\nu$)
-pag 6. "large-scale generative models" doesn't seem particularly accurate for the shown experiments (32x32 to 64x64 images).




**Summary Of The Paper:**

This work introduces a method for doing generative modeling, and conditional generative modeling (e.g., image restoration) using the optimal transport map between two distributions.

The paper first discusses the different uses of Optimal Transport (OT) on generative models, and shows that most of the works are on using the OT cost and not the map/plan itself. The paper also discusses some of the work that does show the OT map but only into latent spaces due to the intrinsic complexity of computing the OT map on the original space.

The proposed method uses the Wasserstein-2  distance as a cost measure and assumes the existence of a unique OT deterministic plan, implying the existence of a map.  The formulation of the OT plan estimation problem (Lemma 4.1) ends up addressing a saddle point problem (max-min).

The analysis requires that the dimensions of input and output spaces be the same. The method also proposes to use a deterministic and non-learnable mapping to embed the input signals on a space that has the same dimension of the output space, so the framework can be applied. This embedding needs to be manually defined depending on the application.

The paper presents experiments on image generation and image restoration on classical  datasets: MNIST, CIFRAR, CelebA (64x64). Regarding image restoration three different applications are analyzed: denoising, colorization inpainting.

The method compares similarly to other generative modeling models (in particular similar to WGANs).


**Summary Of The Review:**

This paper introduces a mathematical formulation to recover the OT map that maps two distributions for doing generative modeling and (unpaired) image restoration tasks. The main weaknesses of the paper are the presentation and experimental validating the approach.  It's hard to understand what is different with respect to previous work also doing OT map estimation. What's particularly interesting about this mathematical formulation that enables it to work on the original dimension?  Experiments are only on small images which implies that the claim of working on high-dimensions and large-scale datasets is a little too strong.

--
After more discussion with the authors and after considering the updated manuscript I'm raising my score (marginally above the acceptance threshold).

---

> ### Author Response · Authors · 2021-11-15
> **Answer to Reviewer qNeK (Part 1/2)**
>
> Dear reviewer qNeK,
>
> Thanks for your thoughtful review. Please find below the answers to your comments and questions.
>
> **(1) Long introduction**
>
> Since our proposed learning pipeline (OT map as a generative model) differs from previously existing OT-based approaches, we think it is essential to clearly discuss the differences. Therefore, in Section 3 we provide a comprehensive overview of existing alternative OT-based approaches.
>
> **(2) Lemma 4.1 and connection to the previous paragraphs.**
>
> We added additional explanations to Section 4.1 to make the flow of the paper more transparent.
>
> **(3) Have people tried the idea [of our lemma 4.1] of estimating an OT map before? Why the proposed formulation should work better than the other methods when addressing the problem in the original dimension?**
>
> At the end of Section 3, we refer to the recent evaluation of OT map methods for $\mathbb{W}_2$ (Korotin et al., 2021b) which shows that most existing methods for $\mathbb{W}_2$ between **equal dimensions** are poorly scalable. This is primarily due to poor expressiveness of input-convex neural networks (ICNNs -- a necessary parametrization in some methods) or bias. Therefore, we do not provide a detailed overview of these methods in our paper.
>
> Additionally, after Lemma 4.1 we discuss the methods which use an idea similar to ours. The closest related method is $\lfloor\text{MM:R}\rceil$ (Korotin et al., 2021b) which neither suffers from bias nor requires using ICNN parametrization, therefore presumably scales well. The authors evaluated the method in recovering the transport map in their **artificial benchmark pairs** (their Sections 4.3, 4.4) and as the **loss in GAN** (their Section 4.5). The method has not been tested as a generative model directly on real-world data.
>
> The difference between our method and $\lfloor\text{MM:R}\rceil$ is in parametrization -- they parametrize $\frac{1}{2}\|\cdot\|^{2}-\psi(\cdot)$ with a neural network while we directly parametrize $\psi(\cdot)$. We added additional evaluation of our method to Appendix B.2 on the continuous Wasserstein-2 benchmark (Korotin et al., 2021b).
>
> Our key contribution is an approach that allows using the method between spaces of **unequal dimension** supported by the theoretical guarantees. Presumably, the same methodology with $Q$-embedding can be applied for other OT methods which use the insights of Lemma 4.1, e.g., $\lfloor\text{MM:R}\rceil$. We state this in the conclusion section.
>
> **(3) Non-deterministic optimal plans in unpaired image restoration**
>
> We agree that the optimal plan $\pi^*$ in image restoration task might be non-deterministic. Nevetheless, when $\mu$ and $\nu$ are supported on the compact subsets of $\mathbb{R}^{D}$ and $\mu$ is atomless, for every $\epsilon>0$ there always exists a  1-to-1 map $T^\epsilon_{\#}\mu=\nu$ which provides $\epsilon$ sub-optimal quadratic cost (Santambrogio, 2015, Theorem 1.32), i.e. $$\int\frac{1}{2}\|T^{\epsilon}(x)-x\|^{2}d\mu(x)\leq \mathbb{W}_{2}^{2}(\mu,\nu)+\epsilon.$$
>
> Importantly, there always exists a neural network that approximates $T^{\epsilon}$ with an arbitrarily small error in $\mathcal{L}^{2}(\mu)$ norm. This is due to neural networks being a dense subset in $\mathcal{L}^{2}(\mu)$ space, see our explanations to Reviewer fFFr. Thus, theoretically, learning non-deterministic plans with deterministic neural networks is possible.
>
> In practice, we did not experience issues with learning deterministic maps in unpaired restoration. Note that various other (non OT-based) paired/unpaired image restoration methods also fit a deterministic map, see, e.g., (Yuan et al., 2018) or (Cheng et al., 2015). Extending our method to learning non-1-to-1 mappings is a promising avenue for future work.
>
> **(4) Large-scale experiments.**
>
> The majority of existing work for computing OT maps consider latent/feature spaces with small dimension, e.g., $D=128$. Therefore, w.r.t. those works, our method considers large scale distributions at higher dimensions, e.g., $D=12288$.
>
> To further address your comment about the scalability, during the rebuttal period we tested our method in Celeba $128\times 128$ faces generation. The qualitative results are shown in Figure 11 of Appendix B.3 in the updated paper.
>
> **Concluding remarks.** Please respond to our post to let us know if the clarifications above suitably address your concerns about our work. We are happy to address any remaining points during the discussion phase; if the responses above are sufficient, we kindly ask that you consider raising your score.

---

> > ### Author Response · Authors · 2021-11-15
> > **Answer to Reviewer qNeK (Part 2/2)**
> >
> > **References**
> >
> > Santambrogio, F. (2015). Optimal transport for applied mathematicians. Birkäuser, NY, 55(58-63), 94.
> >
> > Yuan, Y., Liu, S., Zhang, J., Zhang, Y., Dong, C., \& Lin, L. (2018). Unsupervised image super-resolution using cycle-in-cycle generative adversarial networks. In Proceedings of the IEEE Conference on Computer Vision and Pattern Recognition Workshops (pp. 701-710).
> >
> > Cheng, Z., Yang, Q., \& Sheng, B. (2015). Deep colorization. In Proceedings of the IEEE International Conference on Computer Vision (pp. 415-423).
> >
> > Korotin, A., Li, L., Genevay, A., Solomon, J., Filippov, A., \& Burnaev, E. (2021b). Do Neural Optimal Transport Solvers Work? A Continuous Wasserstein-2 Benchmark. NeurIPS 2021.

---

> > > ### Author Response · Authors · 2021-11-28
> > > **Looking forward to your final feedback**
> > >
> > > Dear reviewer qNeK,
> > >
> > > We thank you for your review and appreciate your time reviewing our paper.
> > >
> > > The end of the discussion phase is approaching. We would be grateful if we could hear your feedback regarding our revision and answers to the reviews. We are happy to address any remaining points during the remaining period.
> > >
> > > Thanks in advance,
> > >
> > > the authors of "Generative Modeling with Optimal Transport Maps"

---

> > ### Comment · Reviewer_qNeK · 2021-11-29
> > **Final comments.**
> >
> > I thank the authors for their response and the additional information/experiments. The updated version of the paper, with a little more information here and there, improves the readability and the flow. I still think that the introduction, and the paper itself could go to the point earlier in the manuscript so the reader understands the key contributions from the start (need to wait till the middle of Section 4 to understand what the paper real contributions are).
> >
> > Having said this, my main concern is about the application in question to restore images. It is evident that in image restoration tasks the OT-problem (generally) does admit a deterministic solution (since it is ill-posed). I would have liked this to be discussed in much more depth in the paper, in particular for example taking a concrete case, image colorization. It's unclear how one of the infinite possible solutions is obtained and chosen (through the deterministic map). This gives the impression that the application to image restoration is a bit forced, and not carefully analyzed (i.e., a big gap between theory and practice). But I understand that the idea is interesting and can contribute future work to develop more in this sense so I'm raising my score to the minimum recommendation for acceptance.

---

### Official Review · Reviewer_vEgs · 2021-11-07

**Correctness:** 2
**Technical Novelty And Significance:** 2
**Empirical Novelty And Significance:** 2
**Recommendation:** 5
**Confidence:** 4

**Main Review:**

Pros:
- The writing is clear and easy to follow.
- The experimental results are solid and convincing.

Cons:
- I am not sure if equation (10) and equation (11) is really equivalent to each other. To apply the Brenier theorem, namely replace $y$ with $T(x)$ in equation (10), we need to add the measure preserving constraint $T_\#\mu = \nu$ in equation (11). Without this constraint, which requires $T$ to be a measure preserving map, equation (11) may goes to $\infty$. For example, $T(x)\equiv 0$ and $\psi(y)=-\infty$. The optimal transport map should be included in the feasible set of equation (11) if it is large enough, but without the measure preserving constraint, it's hard to prove that the learned map $T$ is measure preserving. Therefore, the authors need to make some effort to show the equivalence.

- Furthermore, to make the conclusion more convincing, I also recommend the authors to experiment on more 2-dimensional toy sets. If the learned map $T$ is an optimal transport map, the samples of the Gaussian distribution should be evenly transported to the Mixture of 8 Gaussians in Fig. 14(f). But it is really difficult to find the 'even' transportation (especially for the southeast samples), which weakens the conclusion that the learned is an optimal transport map.

**Summary Of The Paper:**

The paper proposes to learn the optimal transport map from the latent distribution to the data distribution directly by optimizing the W2 distance. To find the OT map, the authors replace $y$ with $T(x)$ in the Kantorovich dual problem.
Experiments show that the proposed method works well and learns the image distribution successfully.



**Summary Of The Review:**

My main concern comes from the deduction from equation (10) to equation (11). The authors may add more explanation to show the equivalence.

---

> ### Author Response · Authors · 2021-11-15
> **Answer to Reviewer vEgs**
>
> Dear reviewer vEgs,
>
> Thanks for your thoughtful review. We are encouraged since you found our writing clear and easy to follow, and the experimental results are solid and convincing. Please find below the answers to your comments and questions.
>
> **(1) Transition between equations (10) and (11).**
>
> We think that there is a misunderstanding in the transition from equation (10) to (11). Their equivalence follows from the interchange between supremum and integral provided by the **Rockafellar's interchange theorem**, see (Rockafellar, 1976, Theorem 3A). According to the theorem, under mild assumptions on a function $f$ on $\mathcal{X}\times\mathcal{Y}$ and a distribution $\mu$ on $\mathcal{X}$, it holds
> $$\int_{\mathcal{X}}\sup_{y\in\mathcal{Y}}f(x,y)d\mu(x)=\sup_{T:\mathcal{X}\rightarrow\mathcal{Y}}\int_{\mathcal{X}}f(x,T(x))d\mu(x).$$
> Note the theorem's original formulation uses infimum instead of supremum, but it is analogous.
>
> In our case, we derive equation (11) from (10) using the interchange in the Rockafellar's theorem for $f(x,y)=\langle x,y\rangle-\psi(y)$. Please note the interchange is not related to the Brenier's theorem; the measure-preserving condition here is not relevant. However, when $\psi=\psi^*$ is an optimal potential, our Lemma 4.1 states that the optimal map $T^*$ is contained in the argsup set of equation (11). The optimal map $T^*$ is measure-preserving by its definition.
>
> Following your suggestion, we extended the explanations related to equations (10) and (11).
>
> **(2) Are the computed maps optimal?**
>
> Following your suggestion, we added several additional qualitative 2D examples to Appendix B.3 (Figure 18) to show that the maps computed by our method are optimal.
>
> To further address your comment, we conducted the quantitative evaluation of our method on the recent Wasserstein-2 benchmark (Korotin et al., 2021b) in Appendix B.2. The benchmark provides pairs of continuous distributions with analytically known OT maps between them. For evaluation, we used their high-dimensional "Early" images benchmark pair (dimension $D=12288$), see subsection 4.1 and Figure 4a of the benchmark paper for details. Our evaluation (Table 4) shows that for our method the $\mathcal{L}_{2}$-UVP$\downarrow$ metric (which the authors of the benchmark use in their evaluation) is only  $\approx 1\%$. Qualitatively, the recovered maps are also good, see Figure 10.
>
>
> **Concluding remarks.** Please respond to our post to let us know if the clarifications above suitably address your concerns about our work. We are happy to address any remaining points during the discussion phase; if the responses above are sufficient, we kindly ask that you consider raising your score.
>
> **References**
>
> Rockafellar, R. T. (1976). Integral functionals, normal integrands and measurable selections. In Nonlinear operators and the calculus of variations (pp. 157-207). Springer, Berlin, Heidelberg.
>
> Korotin, A., Li, L., Genevay, A., Solomon, J., Filippov, A., \& Burnaev, E. (2021b). Do Neural Optimal Transport Solvers Work? A Continuous Wasserstein-2 Benchmark. NeurIPS 2021.

---

> > ### Comment · Reviewer_vEgs · 2021-11-27
> > **More comments**
> >
> > I thank the authors for the thorough responses. The additional experiments in Fig. 18 are promising.
> > But I am still skeptical about the equivalence between equation (10) and (11). If we replace $y$ sampled from $\nu$ in equation (10) with $y=T(x)$, where $x$ is sampled from $\mu$. Then the inherent requirement of $T$ is $T_\mu = \nu$. Thus, equation (10) should be equivalent to
> >
> > $const - \inf_\psi \sup_{T_\mu=\nu}\int_X \left( \langle x, T(x)\rangle - \psi(T(x) \right) d\mu(x) +\int_Y \psi(y)d\nu(y)~~~~~ (a)$
> >
> > By removing the measure preserving requirement of $T$ in the above equation (a), we finally get the equation (11), which decouples the relationship between $\mu$ and $\nu$. The searching space of the above equation (a) and that of (11) is different. In theory, the searching space of equation (a) should be a subset of (11). I am not sure if equation (a) and equation (11) is really equivalent to each other. If it is, the equivalence should be proven.

---

> > > ### Author Response · Authors · 2021-11-28
> > > **The interchange and the measure-preserving constraint**
> > >
> > > Dear reviewer vEgs,
> > >
> > > Thanks for considering the revision and our answers. We are glad that you are positive about our newly added experiments showing that
> > > our method indeed **recovers OT maps**.
> > >
> > > However, we insist that our transition from (10) to (11) is valid and the provided explanations in the paper are rigorous and sufficient. Please consider our replies to your comments below.
> > >
> > > **(1) If we replace $y$ sampled from $\nu$ in equation (10) with $y=T(x)$, where $x$ is sampled from $\nu$, then the inherent requirement of $T$ is $T_{\#}\mu=\nu$.**
> > >
> > > In transition from (10) to (11), we work **exclusively** with the integral
> > > $\int_{\mathcal{X}}\sup_{y\in\mathbb{R}^{D}}\left\lbrace\langle x,y\rangle-\psi(y)\right\rbrace d\mu(x).$
> > > To compute it, one needs to sample $x\sim\mu$ and substitute the respective $y^*=y^*(x)$ which is a solution of the optimization problem $\sup_{y\in\mathbb{R}^{D}}\left\lbrace\langle x,y\rangle-\psi(y)\right\rbrace$. The integral is **completely independent** of measure $\nu$. Consequently, when we apply the Rockafellar's interchange theorem to introduce $T$, **no constraints** on $T$ **related to** $\nu$ **appear**; the integral equals
> > > $\sup_{T:\mathcal{X}\rightarrow\mathcal{Y}}\int_{\mathcal{X}}\left\lbrace\langle x,T(x)\rangle-\psi\big(T(x)\big)\right\rbrace d\mu(x).$
> > >
> > > **(2) By removing the measure-preserving requirement of $T$ in the above equation (a), we finally get the equation (11), which decouples the relationship between $\mu$ and $\nu$.**
> > >
> > > The relationship between $\mu$ and $\nu$ is not decoupled. It is **maintained** through the outer optimization (infimum) over $\psi$, The function $\psi$ appears in both the above mentioned integral and the second term $\int_{\mathcal{Y}} \psi(y) d\nu(y)$.
> > >
> > > **Concluding remarks.** To further alleviate your concerns, we additionally emphasize that derivations related to our transition between (10) and (11) appear in other OT publications [1, Equation 8], [2, Theorem 3.3], [3, Equation 10], [4, Equation 9] which we discuss after Lemma 4.1. We hope that our additional clarifications sufficiently address your concerns and kindly ask you to consider raising the score for our paper.
> > >
> > > **References**
> > >
> > > [1] Nhan Dam, Q. H., Le, T., Nguyen, T. D., Bui, H., \& Phung, D. (2019). Three player Wasserstein GAN via amortised duality. In Proc. of the 28th Int. Joint Conf. on Artificial Intelligence (IJCAI).
> > >
> > > [2] Makkuva, A., Taghvaei, A., Oh, S., \& Lee, J. (2020, November). Optimal transport mapping via input convex neural networks. In International Conference on Machine Learning (pp. 6672-6681). PMLR.
> > >
> > > [3] Korotin, A., Egiazarian, V., Asadulaev, A., Safin, A., \& Burnaev, E. (2020, September). Wasserstein-2 Generative Networks. In International Conference on Learning Representations.
> > >
> > > [4] Korotin, A., Li, L., Genevay, A., Solomon, J., Filippov, A., \& Burnaev, E. (2021). Do Neural Optimal Transport Solvers Work? A Continuous Wasserstein-2 Benchmark. arXiv preprint arXiv:2106.01954. (NeurIPS 2021)

---

> > > > ### Comment · Reviewer_vEgs · 2021-11-29
> > > > **Additional comments**
> > > >
> > > > I thank the authors for the further responses. However, I am still skeptical about the equivalence. The authors claim that the above references use the same strategy to show the equivalence between equation (10) and (11), where I checked [2] and [3] in detail.
> > > >
> > > > In [2], the transport map $T$ is given by the gradient of the input convex neural networks (ICNN), which is used to represent the convex Brenier potential.
> > > >
> > > > In [3], $\psi$ is required to be convex.
> > > >
> > > > In the both papers, to apply the Brenier theorem which states that the optimal transport map $T$ should be the gradient of the convex Brenier potential, they choose to parameterize either $\phi$ ($T=\nabla \phi$) or $\psi$ by ICNNs, namely requiring $\phi$ or $\psi$ to be convex. However, in equation (11), there is no constraints about $T$ or $\psi$. Thus, I think the authors should prove the equivalence between equation (10) and (11).

---

> > > > > ### Author Response · Authors · 2021-11-29
> > > > > **Further discussion on equivalence**
> > > > >
> > > > > Dear reviewer vEgs,
> > > > >
> > > > > Thanks for considering our answers. We note that in derivations of equations (10) and (11) **the Brenier's theorem is not used**. The equivalence follows from the equality of $\mathcal{L}_1 := \int_\mathcal{X} \sup\_{y\in \mathcal{Y}} \big[ \left \langle x, y \right \rangle - \psi (y)\big]d\mu(x)$ and $\mathcal{L}_2 := \sup\_{T: \mathcal{X}\rightarrow \mathcal{Y}} \big[\int_\mathcal{X} \left \langle x, T(x) \right \rangle - \psi\left ( T\left ( x \right ) \right )d\mu(x)\big]$, i.e, the inner integrals of equations (10) and (11) respectively. This follows from **the Rockafellar's interchange theorem**. We also provide an independent proof below.
> > > > >
> > > > > **Proof.** Pick any $T:\mathcal{X}\rightarrow\mathcal{Y}$. For every point $x\in\mathcal{X}$ by the definition of the supremum we have
> > > > > $$\langle x, T(x) \rangle - \psi\left (T(x )\right)   \leq \sup_{y\in\mathcal{Y}}\left\lbrace\langle x, y \rangle - \psi(y)\right\rbrace.$$
> > > > > Integrating the expression w.r.t. $x\sim\mu$ yields
> > > > > $$\int_\mathcal{X}\lbrace\langle x, T(x) \rangle - \psi\left (T(x )\right)\rbrace d\mu(x)   \leq \int_\mathcal{X}\sup_{y\in\mathcal{Y}}\left\lbrace\langle x, y \rangle - \psi(y)\right\rbrace d\mu(x)=\mathcal{L}_{1}.$$
> > > > > Since the inequality holds for all $T:\mathcal{X}\rightarrow\mathcal{Y}$, we conclude that
> > > > > $$\mathcal{L}_2=\sup\_{T:\mathcal{X}\rightarrow\mathcal{Y}}\int_\mathcal{X}\lbrace\langle x, T(x) \rangle - \psi\left (T(x )\right)\rbrace d\mu(x)   \leq \int_\mathcal{X}\sup\_{y\in\mathcal{Y}}\left\lbrace\langle x, y \rangle - \psi(y)\right\rbrace d\mu(x)=\mathcal{L}_1,$$
> > > > > i.e. $\mathcal{L}_2\leq \mathcal{L}_1$. Now let us prove that the sup on the left side actually equals $\mathcal{L}_1$. To do this, we need to show that for every $\epsilon>0$ there exists $T^{\epsilon}:\mathcal{X}\rightarrow\mathcal{Y}$ satisfying
> > > > > $$\int_\mathcal{X}\lbrace\langle x, T^{\epsilon}(x) \rangle - \psi\left (T^{\epsilon}(x)\right)\rbrace d\mu(x)\geq \mathcal{L}_1-\epsilon.$$
> > > > > First note that for every $x\in\mathcal{X}$ by the definition of the supremum there exists $y^\epsilon=y^\epsilon(x)$ which provides
> > > > > $$\langle x, y^\epsilon(x) \rangle - \psi\left (y^\epsilon(x)\right)   \geq \sup\_{y\in\mathcal{Y}}\left\lbrace\langle x, y \rangle - \psi(y)\right\rbrace-\epsilon.$$
> > > > > We take $T^\epsilon(x)=y^\epsilon(x)$ for all $x\in\mathcal{X}$ and integrate the previous inequality w.r.t. $x\sim\mathcal{\mu}$. We obtain
> > > > > $$\int_\mathcal{X}\lbrace{\langle x, T^\epsilon(x) \rangle - \psi\left (T^\epsilon(x)\right)\rbrace}d\mu(x)   \geq \int_\mathcal{X}\sup\_{y\in\mathcal{Y}}\left\lbrace\langle x, y \rangle - \psi(y)\right\rbrace d\mu(x) -\epsilon=\mathcal{L}_1-\epsilon,$$
> > > > > which is the desired inequality. Q.E.D.
> > > > >
> > > > > **Concluding remarks.**  Please note that the papers [2] and [3] use the Brenier's theorem only to restrict optimization over $\psi$, $T$ (in our notation) to convex $\psi$ and $T=\nabla\phi$ for some convex $\phi$. The papers [1] and [4] consider unconstrained formulations. We hope that our additional clarifications sufficiently address your concerns and kindly ask you to consider raising the score for our paper.

---

### Author Response · Authors · 2021-11-15
**Please consider the updated version**

Dear reviewers,

We thank you for spending time reviewing our paper. Please find the answers to your questions in our replies to your reviews. We have incorporated your suggestions into the paper (the changes are highlighted with the blue color) and uploaded the updated version. The major changes are:

(1) We added Figure 4 to Section 4.2 with the **scheme of our method**;

(2) We added additional qualitative results in **2D examples** (Figure 18 in Appendix B.3);

(3) We added quantitative and qualitative evaluation of our method on the Wasserstein-2 benchmark to show that it **recovers OT maps well** (Appendix B.2, Table 4 and Figure 10);

(4) We added additional experiments in generative modeling of **Celeba 128x128** (Figure 11).

Please consider the updated version and respond to our replies to let us know if the clarifications and additions suitably address your concerns about our work.  We are happy to address any remaining points during the discussion phase.

---

### Author Response · Authors · 2021-11-20
**The current stage of the discussion is ending soon**

Dear Reviewers,

The end of the current stage of the discussion period is approaching. In the next stage, it will not be possible to upload an updated revision. We would be grateful if we could hear your feedback regarding our revision and answers to the reviews. We are happy to address any follow-up points during the remaining discussion period and update the paper accordingly.

Thanks in advance,

the authors of "Generative Modeling with Optimal Transport Maps"

---

### Author Response · Authors · 2021-12-05
**Optimality of the maps and extra qualitative examples**

Dear area chair and reviewers,

Before the discussion ends, we would like to summarize and share further empirical evidence that our method successfully computes OT maps in computer vision problems.

In the paper, we considered popular image generation and unpaired restoration tasks that already form a representative set of examples. Additionally, we conducted an experiment with **unpaired image-to-image translation** between *HandBags* and *Shoes* datasets from the iGAN [1] repository [2]. For two experiments (*HandBags*$\rightarrow$*Shoes* and *Shoes*$\rightarrow$*Handbags*), we provide qualitative results of translation via the following anonymous GDrive link:

https://drive.google.com/file/d/1D1jiwT6YWnSVEiVmwse5H0nd3H2hpGSO/view?usp=sharing

We show the evolution of the OT map during training. The 1st line of each subplot shows input images, the 2nd line -- their **translation** with OTM, the 3rd line -- images (unpaired) from the output dataset. Our algorithm works well as a generative model. Importantly, the learned map **preserves the texture** of the input images, e.g., *handbags2shoes/18500.png*, where bags are translated to shoes with the same style. This indicates that the map is indeed optimal since it tries to minimally change the input image in the pixel space.

Taking the already provided in the paper and newly presented experiments into account, we think we have provided enough evidence of the practical performance of our OTM algorithm. We kindly ask you to take this into account when making the final decision.

**References**

[1] Zhu, J. Y., Krähenbühl, P., Shechtman, E., \& Efros, A. A. (2016, October). Generative visual manipulation on the natural image manifold. In European conference on computer vision (pp. 597-613). Springer, Cham.

[2] https://github.com/junyanz/iGAN/blob/master/train_dcgan/README.md

---

### Decision · Program_Chairs · 2022-01-20

**Decision:**

Accept (Poster)

**Comment:**

The paper proposes a new method to learn OT maps, and reframes it in the GAN literature. The initial method works when computing maps between equal dimensions, through duality and an identity (10 - 11, amply discussed in the reviewing process). Lemma 4.1 provides the main result. While the discussion right below on the fact that several functions (non-OT maps) might maximize that criterion is not completely satisfactory, the result provides an interesting characterization. The second contribution adds a method to compute OT maps between spaces of unequal dimensions. Overall the contribution sounds a bit ad-hoc, and one wonders whether this does really work (comments such as "we add small gradient penalty (Gulrajani et al., 2017) on potential ψω for better stability. The penalty in not included in Algorithm 1 to keep it simple." are strange and point to instability) but the overall creativity and new ideas in the paper seem to have garnered enough support from reviewers to push for an accept.